# A Comparison of BPNN, GMDH, and ARIMA for Monthly Rainfall Forecasting Based on Wavelet Packet Decomposition

Wenchuan Wang [1],*, Yujin Du [1], Kwokwing Chau [2], Haitao Chen [1], Changjun Liu [3] and Qiang Ma [3]

1 Henan Key Laboratory of Water Resources Conservation and Intensive Utilization in the Yellow River Basin, College of Water Resources, North China University of Water Resources and Electric Power, Zhengzhou 450046, China; 13949121629@163.com (Y.D.); chenghaitao@ncwu.edu.cn (H.C.)
2 Department of Civil and Environmental Engineering, The Hong Kong Polytechnic University, Hung Hom, Kowloon, Hong Kong, China; cekwchau@polyu.edu.hk
3 China Institute of Water Resources and Hydropower Research, Beijing 100081, China; lcj2005@iwhr.com (C.L.); maqiang@iwhr.com (Q.M.)
* Correspondence: wangwenchuan@ncwu.edu.cn

**Abstract:** Accurate rainfall forecasting in watersheds is of indispensable importance for predicting streamflow and flash floods. This paper investigates the accuracy of several forecasting technologies based on Wavelet Packet Decomposition (WPD) in monthly rainfall forecasting. First, WPD decomposes the observed monthly rainfall data into several subcomponents. Then, three data-based models, namely Back-propagation Neural Network (BPNN) model, group method of data handing (GMDH) model, and autoregressive integrated moving average (ARIMA) model, are utilized to complete the prediction of the decomposed monthly rainfall series, respectively. Finally, the ensemble prediction result of the model is formulated by summing the outputs of all submodules. Meanwhile, these six models are employed for benchmark comparison to study the prediction performance of these conjunction methods, which are BPNN, WPD-BPNN, GMDH, WPD-GMDH, ARIMA, and WPD-ARIMA models. The paper takes monthly data from Luoning and Zuoyu stations in Luoyang city of China as the case study. The performance of these conjunction methods is tested by four quantitative indexes. Results show that WPD can efficiently improve the forecasting accuracy and the proposed WPD-BPNN model can achieve better prediction results. It is concluded that the hybrid forecast model is a very efficient tool to improve the accuracy of mid- and long-term rainfall forecasting.

**Keywords:** monthly rainfall forecasting; back-propagation neural network; group method of data handing; autoregressive integrated moving average; wavelet packet decomposition

## 1. Introduction

Hydrological time series forecasting is essential for a variety of real-world managements or operation of water resources systems [1,2]. Precipitation is affected by many factors such as atmospheric circulation, topography, climate change, and human activities. The improvement of precipitation prediction has received a lot of attention across the world and many models have been constructed to improve the hydrological process simulation and prediction accuracy [3–5].

These models can fall into two categories: knowledge-based models and data-based models [6]. Knowledge-based model is a numerical simulation technology that describes natural phenomena on the basis of an internal physical mechanism of the system [7]. However, the lack of multisource information and optimization complexity of computation parameters limit the generalization of physical-based models [8]. In contrast, data-based models can obtain satisfying results by using historical data without involving the physical load within hydrological time series [9,10]. Hence data-based models have received a lot of attention in the hydrological forecasting field. In this paper, we are devoted to verifying several data-based models for monthly rainfall time series forecasting. There

are many data-based models, e.g., artificial neural networks (ANN) [11], genetic programming (GP) [12], support vector machines (SVM) [13], and adaptive neuro-fuzzy inference system (ANFIS) [14].

Box–Jenkins models [15], which are considered as the most comprehensive tool in all statistical methods of time series forecasting, include auto-regressive (AR), moving average (MA), autoregressive moving average (ARMA), ARIMA, and other models. ARIMA is a linear statistical model and normally used to simulate and forecast time series with temporal correlation [16,17]. With the advance of technology such as computers, communication, remote sensing, and geography information systems, the prediction ability of the ARIMA model has been greatly improved [18]. Rahman et al. [19] used Mann–Kendall, Spearman's rho test, and the ARIMA model to analyze and predict rainfall trends in Bangladesh. Mishra et al. [20] compared seasonal ARIMA and ARIMA models for runoff forecasting accuracy in the River Brahmaputra Basin, and the results indicated that ARIMA could provide higher accuracy. Rizeei et al. [21] combined a soil conservation service–curve number (SCS-CN) model with an ARIMA and land transformation model to monitor the changes of surface runoff. Wang et al. [22] proposed a hybrid Empirical Mode Decomposition (EMD)/Ensemble Empirical Mode Decomposition (EEMD)-ARIMA model for long-term runoff forecasting.

ANN, a nonlinear data-based model, is extensively used for hydrological applications [23,24]. The major application of ANN can be summarized as streamflow forecasting [25,26], rainfall forecasting [27,28], groundwater problems [29,30], suspended sediment estimation [31], regional drought analysis and forecasting [32,33], etc. Among different kinds of ANN, BPNN is a multilayer feed forward ANN with unidirectional transmission, which has advantages of learning and extracting the features, memory association, parallel architecture, and independent learning and adaptive capabilities [34]. The BPNN model has been widely applied to precipitation prediction, study of rainfall prediction with meteorological parameters [35], estimation of regional surface soil moisture [36,37], etc. Consequently, we attempt to use BPNN for monthly rainfall forecasting as a nonlinear data-based model.

GMDH is a sub-model of ANN for complex system modeling [38]. The main principle is to construct an analytic function of the system by quadratic node transfer function. The coefficients of binomial transfer function are obtained by polynomial regression. GMDH has been successfully used in broad fields such as economics, engineering, science, medical diagnostics, control systems, signal processing, and water resources [39,40].

Although the performance of ANN is remarkable in dealing with linear problems, it cannot handle non-stationary and nonlinear problems that arise in rainfall data. Studies have shown that forecasting accuracy of models could be improved by appropriate data preprocessing techniques to eliminate noises in hydrological time series. In recent years, many scholars have performed a lot of work based on this idea to improve the prediction performance of models. Partal and Kişi [41] proposed a wavelet-neuro-fuzzy model, especially suitable for forecasting daily rainfall time series, which have zero rainfall in summer months. Wang, et al. [42] proposed the EEMD-ANN model to forecast medium- and long-term runoff time series. Yu, et al. [43] explored Fourier transform (FT) and support vector regression (SVR) for forecasting monthly reservoir inflow and compared them with EEMD-SVR and SSA-SVR models, and found that FT-SVR consumed more computational resources in parameter calibration. The least-squares wavelet analysis (LSWA) [2] has shown promising results in successful analysis of streamflow and climate time series. Feng, et al. [44] combined variational mode decomposition (VMD), SVM, and quantum-behaved particle swarm optimization (QPSO) to forecast monthly streamflow and achieved excellent prediction results.

Most common decomposition approaches perform well only when the input variables meet certain conditions. For example, EMD may suffer from mode mixing due to intermittent signal [45], and this effect is important to hydrological applications. The stationarity of time series has a great influence on the accuracy of position in the domain identified by

FT method [46]. Nevertheless, hydrological time series are non-stationary, which means that statistical properties will fluctuate over time [47]. In recent years, researchers have paid great attention in WPD. The main idea of WPD method is using multiple filters to decompose the original signal into more linear sub-signals with different frequency characteristics, which can be regarded as an improved version of the wavelet decomposition (WD). In discrete wavelet transform (DWT), when performing next layer decomposition, only approximate coefficients obtained from the upper layer can pass through the filter [48]. However, when the WPD method performs the next level of decomposition, both the low-frequency sequence and high-frequency sequence can pass the filter [49], and the total number of coefficients is still the same without redundancy. Therefore, WPD can extract the features of the original signal more comprehensively, which not only provides a wide range of possibilities for signal analysis but also allows the best matching analysis of the signal. Meanwhile, compared with DWT, the decomposition structure of WPD provides more opportunities to improve computational efficiency [50]. Therefore, WPD is preferred in this paper in consideration of the complex nonlinearity and non-stationary characteristics of hydrologic time series.

The purpose of this paper is to investigate the accuracy of ARIMA, GMDH, and BPNN models based on WPD in monthly rainfall forecasting. Most former research often improve the accuracy of prediction models by optimizing model parameters using optimization algorithms, and the improvement effect of this method is often not obvious. In this paper, the data preprocessing method is adopted to improve the accuracy of forecasting models, which can attain more linear sub-series and significantly reduce the difficulty of prediction. Firstly, we use WPD to decompose original monthly rainfall series into a series of sub-series with different frequencies and spatiotemporal resolutions. Then, the subseries decomposed by WPD are used as input data of ARIMA, GMDH, and BPNN to train for prediction. Finally, the prediction results of each hybrid model are obtained by linearly accumulating the outputs of each submodule.

The paper is arranged as follows: Section 2 introduces the basic theory principles of methods and evaluations indices. The forecasting experiments and discussion are presented in Section 3. Finally, Section 4 concludes the paper.

## 2. Materials and Methods

### 2.1. Study Region

Two hydrological stations located in Yiluo River Basin on the south bank of the middle stream of Yellow River are considered as the case study. Yiluo River is an important first-level tributary of Yellow River and one of the main sources of floods in the lower reaches of Yellow River, with a drainage area of 18,881 km$^2$. The mainstream Luo River is 446.9 km long, and the tributary Yi River is 264.8 km long. Luoning and Zuoyu Stations are located in the middle stream of Luo River and the middle and upper stream of Yi River, respectively. The average annual rainfall of the two stations are 635.2 mm and 834.3 mm, respectively. The inter-seasonal fluctuations of rainfall in two stations are very strong. For Luoning station, the average annual rainfall in December and January are 7.9 mm and 7.5 mm, respectively, and the average annual rainfall in July and August are 115.6 mm and 96.6 mm, respectively, indicating high difficulty of modeling. The location of the study area is shown in Figure 1.

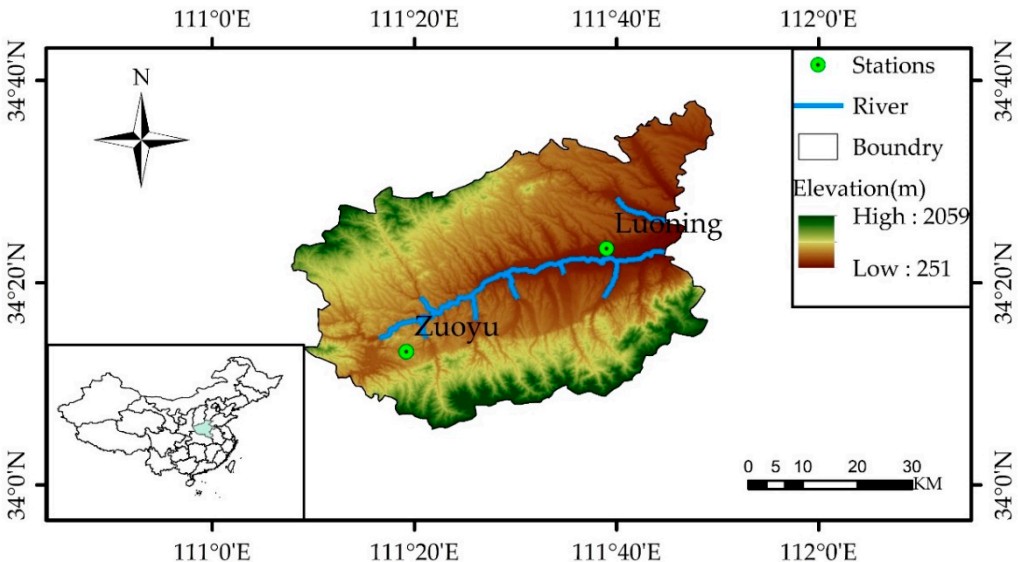

**Figure 1.** Location of Luoning and Zuoyu stations.

### 2.2. Data Sets and Pre-Processing

Monthly rainfall data from two stations are used to investigate the accuracy of several prediction methods. Table 1 shows statistical parameters of monthly rainfall data at Luoning and Zuoyu stations. It can be observed that the original data shows obvious standard deviation, indicating a high difficulty of modeling. Figures 2 and 3 present rainfall data for the two stations, where the data run from 1980 to 2016. In this study, data from 1980 to 2013 are used for training and the final three years are utilized for testing.

**Table 1.** Statistical parameters of monthly rainfall data at Luoning and Zuoyu stations.

| Station | | Max (mm) | Min (mm) | Mean (mm) | Std (mm) |
|---|---|---|---|---|---|
| Luoning | All | 313.8 | 0 | 47.18 | 50.93 |
| | Training | 313.8 | 0 | 47.05 | 50.86 |
| | Testing | 261.7 | 0.4 | 48.66 | 52.39 |
| Zuoyu | All | 430.2 | 0 | 69.53 | 74.39 |
| | Training | 430.2 | 0 | 69.92 | 75.33 |
| | Testing | 316.2 | 0 | 65.05 | 63.44 |

Note: Max is the maximum, Min is the minimum, Std is the standard deviation.

WPD is used to decompose two observed monthly rainfall series into a series of sub-series. The data of all series are divided into training and testing datasets that are normalized to a range of [0, 1] as

$$x_i' = \frac{x_i - \min_{1 \le i \le n}\{x_i\}}{\max_{1 \le i \le n}\{x_i\} - \min_{1 \le i \le n}\{x_i\}} \tag{1}$$

where $x_i'$ and $x_i$ are the normalized and the observed value of the i-th data sample, respectively.

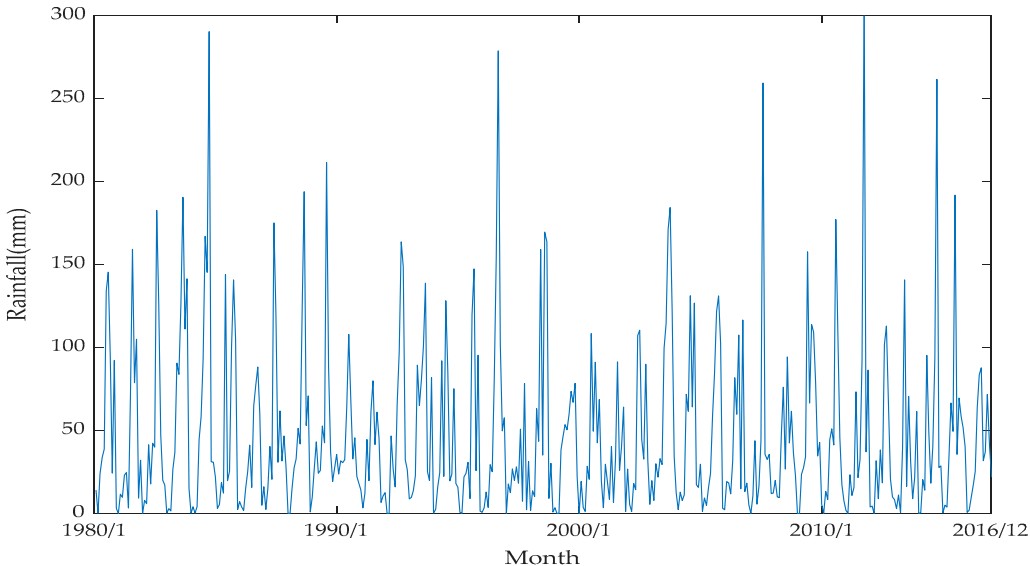

**Figure 2.** Monthly rainfall time series at Luoning station.

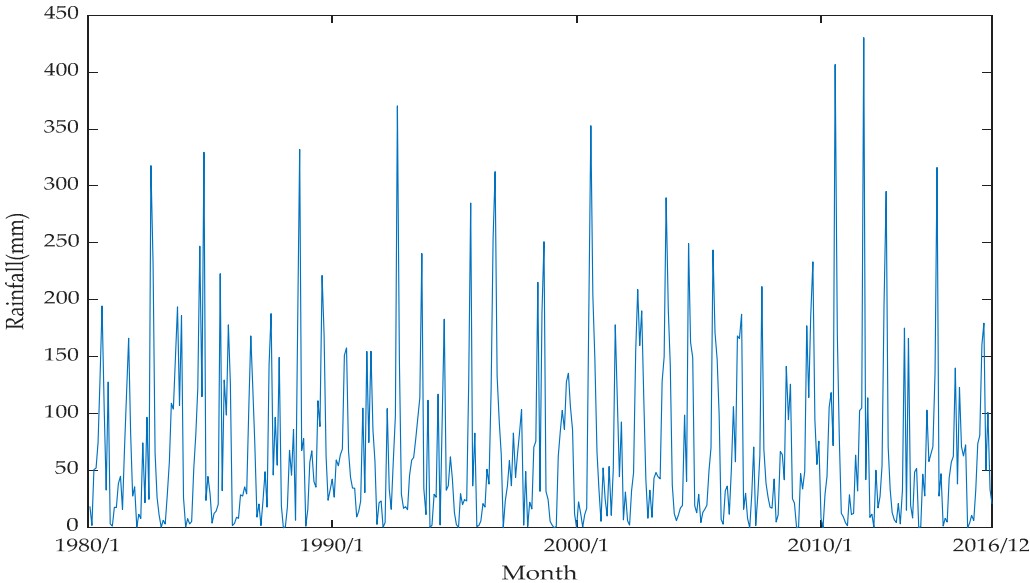

**Figure 3.** Monthly rainfall time series at Zuoyu station.

*2.3. Methods*

2.3.1. ARIMA Model

The ARIMA model proposed by Box and Jenkins [15] has been extensively utilized for analyzing and forecasting hydrologic time series [51]. The principle of ARIMA is to use historical times series to find the forecasting noise, so that the data can be processed smoothly, thus solving the random disturbance problem of the series [52]. ARIMA model construction includes six steps: data acquisition, data preprocessing, model identification, model order determination, parameter estimation, and model verification. Two monthly series collected from China are taken as the test cases. Data preprocessing is the test of stationarity of time series. Recently, ACF (autocorrelation function) and PACF (partial autocorrelation function) are generally adopted to test the stationarity of data. In this paper, the Box–Jenkins method is used for model identification, and Bayes information criteria (BIC) method is used for model order determination. In ARIMA ($p$, $d$, $q$), $p$ represents the number of autoregressive terms, $q$ is the number of moving average terms, and $d$ is the order of differential.

AR model of order $p$, which is written as $AR(p)$, can be expressed as follows:

$$x_t = \phi_1 x_{t-1} + \phi_2 x_{t-2} + \cdots + \phi_p x_{t-p} + \lambda_t \tag{2}$$

The $MA(q)$ model is:

$$x_t = \lambda_t - \psi_1 \lambda_{t-1} - \psi_2 \lambda_{t-2} - \cdots - \psi_q \lambda_{t-q} \tag{3}$$

Thus, the expression of ARMA $(p, q)$ is defined below as:

$$x_t = \phi_1 x_{t-1} + \phi_2 x_{t-2} + \cdots + \phi_p x_{t-p} + \lambda_t - \psi_1 \lambda_{t-1} - \psi_2 \lambda_{t-2} - \cdots - \psi_q \lambda_{t-q} \tag{4}$$

The ARIMA model is obtained by the $d$-order difference of the ARMA model. Therefore, the ARIMA $(p, d, q)$ model is:

$$y_t = \phi_1 x_{t-1} + \phi_2 y_{t-2} + \cdots + \phi_p y_{t-p} + \lambda_t - \psi_1 \lambda_{t-1} - \psi_2 \lambda_{t-2} - \cdots - \psi_q \lambda_{t-q} \tag{5}$$

where $x_t$ represents the predicted value of the model at time $t$, $\phi_i$ is model coefficient, $x_{t-j}$ is previous observation, $\psi_i$ is model parameter related to white noise, $\lambda_t$ is white noise process that obeys a normal distribution with zero mean and variance $\sigma^2$, $\lambda_{t-j}$ is previous noise term, and $y_t = \nabla^d x_t, \cdots$ denotes computation according to the above law. Note that $y_t$ can be replaced with $x_t$ only when $d = 0$.

### 2.3.2. BPNN Model

BPNN, proposed by Fausett [53], is a typical multilayer ANN on the basis of error back propagation. BPNN uses the slope reduction method to find the point(s) with minimum error [54]. These three layers, that is, input layer, hidden layer, and output layer, are employed in BPNN (as shown in Figure 4). The signal is input into the network by the input layer and output by the output layer. BPNN adds several layers (one or more layers) of neurons between the input layer and the output layer. These neurons are called hidden layer neurons. They have no direct contact with the outside world, but the change of their state can affect the relationship between input and output. A conventional three-layer BPNN is used to establish the prediction model of monthly precipitation series in this paper. Tan-sigmoid is the transfer function between the output layer and hidden layer, and the nonlinear Levenberg–Marquardt (LM) algorithm is the training function of BPNN.

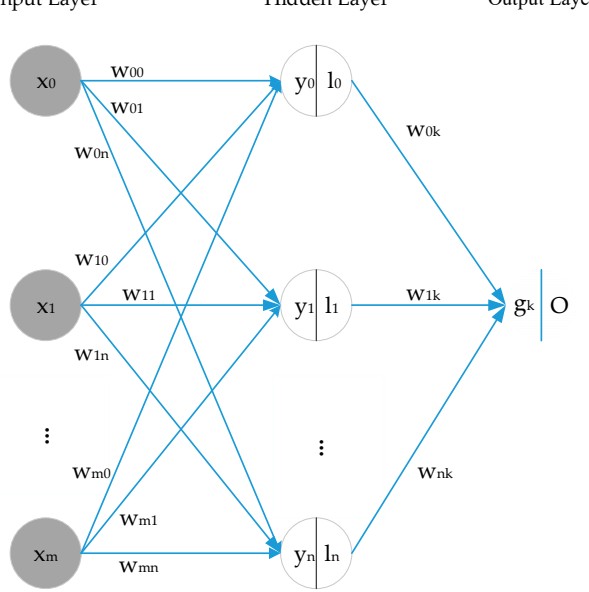

**Figure 4.** Schematic diagram of a BPNN structure.

The mathematical principle of BPNN model is as follows:

$$y_i = \sum_{j=0}^{m} w_{ij}x_j + \beta_j \tag{6}$$

where $x_j$ is input neuron and $j \in (0, m)$, m is the number of input neurons, $w_{ij}$ is weight of the $i$th neuron in the input layer corresponding to the $j$th neuron in the hidden layer, $\beta_j$ is bias-related weight of hidden neurons, $y_i$ is input of the hidden layer node ($i = 0, 1, \ldots, n$), and $n$ is the number of neurons in the hidden layer. Tan-sigmoid is the transfer function between the layer output and the hidden layer, and its form is as follows:

$$l_i = \frac{1}{1 + e^{-y_i}} \tag{7}$$

The output layer is estimated by the following equation:

$$g_k = \sum_{i=0}^{n} w_{ik}l_i + \beta_k \tag{8}$$

$$O = \max(0, g_k) \tag{9}$$

Among them, $g_k$ and $O$ represent input and output values of the output layer, respectively.

The formulas above are the principles of the feedforward propagation mode of the BPNN model. In the process of cyclic simulation, errors generated by the system are collected and returned to the output value. By adjusting the weights and thresholds of neurons, network parameters corresponding to the minimum error are determined to generate an ANN system which can simulate the original problem.

### 2.3.3. GMDH Model

GMDH was developed by Ivakhnenko [55] as a self-organizing approach, which can be applied for multivariate analysis and modeling of complex systems. GMDH has been used to deal with problems of high-order polynomial regression, especially modeling and classification of systems [56]. An important feature of the GMDH method is that external information (i.e., information and data not used in model construction and parameter estimation) is used in modeling, the data of training period is used for modeling, and the information of testing period is only used to select the optimal complexity model. Input and output variables of GMDH are connected by a complex Volterra function in the following form [57]:

$$\hat{y} = s_0 + \sum_{i=1}^{n} s_i x_i + \sum_{i=1}^{n}\sum_{j=1}^{n} s_{ij} x_i x_j + \sum_{i=1}^{n}\sum_{j=1}^{n}\sum_{k=1}^{n} s_{ijk} x_i x_j x_k + \cdots \tag{10}$$

where $x$ denotes the input variable of system, $s_i$ is the weight, $\hat{y}$ is output variable, and $n$ is the number of input variables. Many applications in the quadratic form with only two variables are termed partial descriptions, and use the following form to predict output:

$$\hat{y}_n = s_0 + s_1 x_{ni} + s_2 x_{nj} + s_3 x_{ni}^2 + s_4 x_{nj}^2 + s_5 x_{ni} x_{nj} \tag{11}$$

The coefficient $s_i$ is obtained by minimizing the Mean Square Error (MSE) between the input–output data pairs:

$$\mathrm{minMSE} = \left( \sum_{n=1}^{N} (\hat{y}_n - y_n)^2 \right) / N \tag{12}$$

where $N$ is the sample size of the training set.

The GMDH model adopts the principle of the classic neural network whose signal propagates forward through network nodes. After the weight has been computed, optimal transfer function of the node is obtained and then its output is passed to the next layer of nodes. As shown in Figure 5, the structure of GMDH network is constantly changing during the training process. GMDH will select the input variables that affect the prediction, which means that the connection between neurons in the network is not fixed, but is selected during training to optimize the network structure; The number of layers in the network is also automatically selected to produce maximum accuracy and avoid over fitting. Solid neurons in each layer are selected neurons, and hollow ones represent unselected neurons.

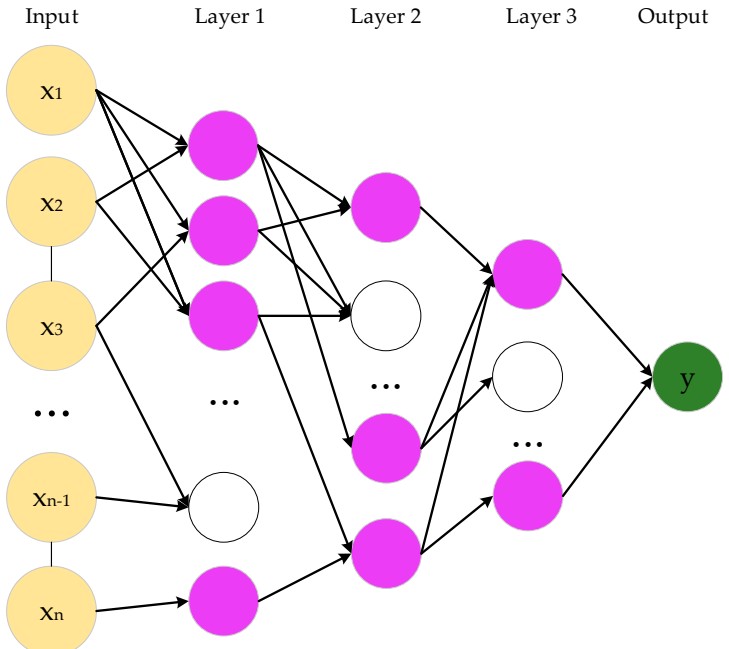

**Figure 5.** Schematic diagram of a GMDH network structure.

2.3.4. Wavelet Packet Decomposition (WPD)

Mallat [58] proposed a Wavelet Representation (WR) theory to compute and interpret multiresolution representation by decomposing the original signal utilizing orthogonal wavelets. WPD reduces the noise of signal by decomposing the signal into different frequencies, which can be regarded as a special WR. In the procedure of orthogonal wavelet decomposition, the signal is decomposed into approximate coefficients and detail coefficients after passing through multiple filters. When performing the next layer decomposition, the upper low-frequency series and high-frequency series are split into two components, and so on. Wavelet function $\psi(t)$ can be defined as:

$$\int_{-\infty}^{+\infty} \psi(t)dt = 0 \tag{13}$$

The form of SWT (successive wavelet transform) of $x$(t) is:

$$W_\psi x(a,b) = |a|^{-1/2} \int_R x(t)\psi * \left(\frac{t-b}{a}\right)dt \tag{14}$$

where $\psi(t)$ represents the mother wavelet and $\psi^*$ denotes its complex conjugate, $a$ and $b(a, b \in R)$ are the scale expansion parameter and time translation parameter, respectively. In engineering applications, input signal is usually discrete. Let $a = a_0^j$, $b = kb_0a_0^j(a_0 > 1, b_0 \in R)$

$(j, n \in Z)$, $j$ and $n$ denote the frequency localization and time localization. The DWT form of $x(t)$ is shown in the following equation:

$$W_\psi x(j,k) = a_0^{-j/2} \int_R x(t) \psi * (a_0^{-j} t - kb_0) dt \tag{15}$$

Normally, we can set $a_0 = 2$ and $b_0 = 1$, and this case is the most efficient for practical applications [58]. Therefore Equation (14) becomes the binary orthogonal wavelet transform:

$$W_\psi x(j,k) = 2^{-j/2} \int_R x(t) \psi * (2^{-j} t - k) dt \tag{16}$$

Unlike DWT, WPD passes more filters, which decompose the signal using both high-frequency components and low-frequency components:

$$\begin{aligned} \phi_{j,k}(t) &= 2^{-j/2} \phi(2^{-j} t - k) \\ \psi_{j,k}(t) &= 2^{-j/2} \psi(2^{-j} t - k). \end{aligned} \tag{17}$$

where $\phi_j(t)$ is the scaling function or the approximation coefficients, and $\psi_j(t)$ is wavelet function (also termed detail coefficient). The two functions correspond to two finite pulse filters, namely, low-pass filter (LPF) $h(n)$ and high-pass filter (HPF) $g(n)$. Hence, the equation of orthogonal wavelet packet is:

$$W_\phi x_{2n}(t) = \sqrt{2} \sum_{k \in Z} h_n \phi_n(2t - k) \tag{18}$$

$$W_\psi x_{2n+1}(t) = \sqrt{2} \sum_{k \in Z} g_n \psi_n(2t - k) \tag{19}$$

where $h(n)$ and $g(n)$ are subject to the following condition:

$$\sum_n h(n)^2 = 1, \sum_n g(n)^2 = 1, \sum_n h(n) = \sqrt{2}, \sum_n g(n) = 0 \tag{20}$$

The wavelet packet function is written by:

$$W_{j,n,k}(t) = 2^{j/2} W_n(2^{-j} t - k) \tag{21}$$

The wavelet packet coefficients can be computed by:

$$W_{j,n,k} = \int x(t) W_{j,n,k}(t) dt \tag{22}$$

Figure 6 illustrates the binary tree of a three-layer WPD. The original signal is shown by $x$ and each node corresponds to a frequency band. LPF and HPF represent low-pass filter and high-pass filter, respectively. The original signal is decomposed into eight subsequences by a three-level WPD. $AAA_3$ and $DDD_3$ represent the lowest frequency and highest frequency, respectively.

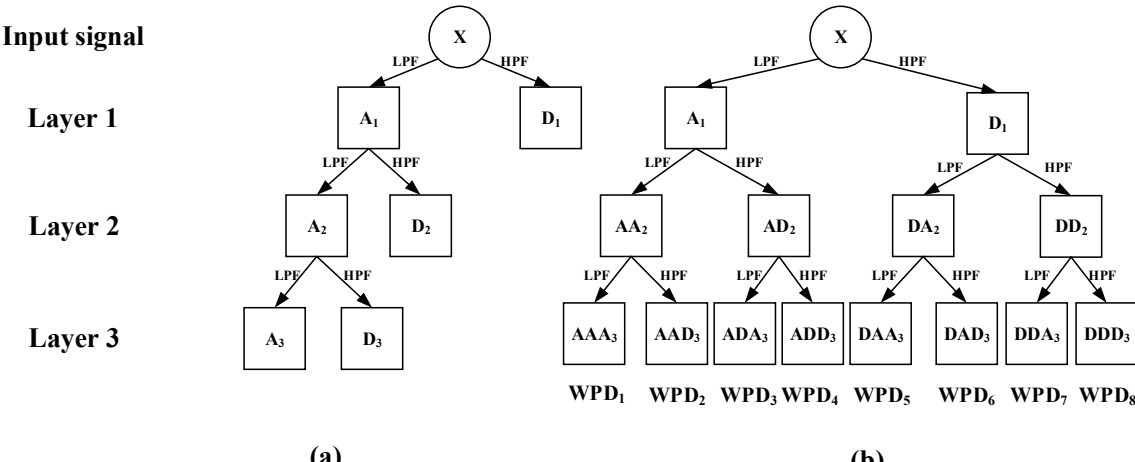

**Figure 6.** Three-layer structure diagram: (**a**) WD; (**b**) WPD.

*2.4. Evaluation Indices*

To evaluate the forecast capacity of different models, four generally adopted standard statistical metrics are used in this study to estimate the global and local errors of models. They are namely RMSE (root mean-squared error) [59], MAE (mean absolute error) [60], R (coefficient of correlation), and NSEC (the Nash–Sutcliffe efficiency coefficient) [61]. RMSE is sensitive even to small errors, which can size the model performance for high rainfall values. However, MAE is suitable for measuring the goodness of fit of model in cases of moderate precipitation. R sizes the degree of collinearity criterion of two variables. NSEC is a widely used index to evaluate the performance measurement of hydrological models. The following formulas are used for computing these parameters:

$$\text{RMSE} = \sqrt{\frac{1}{n}\sum_{i=1}^{n}(Q_y(i) - Q_x(i))^2} \tag{23}$$

$$\text{MAE} = \frac{1}{n}\sum_{i=1}^{n}|Q_y(i) - Q_x(i)| \tag{24}$$

$$R = \frac{\sum_{i=1}^{n}(Q_x(i) - \overline{Q_x})(Q_y(i) - \overline{Q_y})}{\sqrt{\sum_{i=1}^{n}(Q_x(i) - \overline{Q_x})^2 \sum_{i=1}^{n}(Q_y(i) - \overline{Q_y})^2}} \tag{25}$$

$$\text{NSEC} = 1 - \frac{\sum_{i=1}^{n}(Q_y(i) - Q_x(i))^2}{\sum_{i=1}^{n}(Q_x(i) - \overline{Q_x})^2} \tag{26}$$

where $Q_x(i)$ and $Q_y(i)$ are the observed and predicted rainfall, respectively, $\overline{Q_x}$ and $\overline{Q_y}$ represent their average values, and *n* is the total number of input samples.

*2.5. Hybrid Forecasting Models*

This study investigates the accuracy of ARIMA, BPNN, and GMDH models based on WPD in monthly rainfall forecasting. The framework of hybrid models is showed in Figure 7. It can be summarized from Figure 4 that the main steps of the hybrid model prediction architecture are:

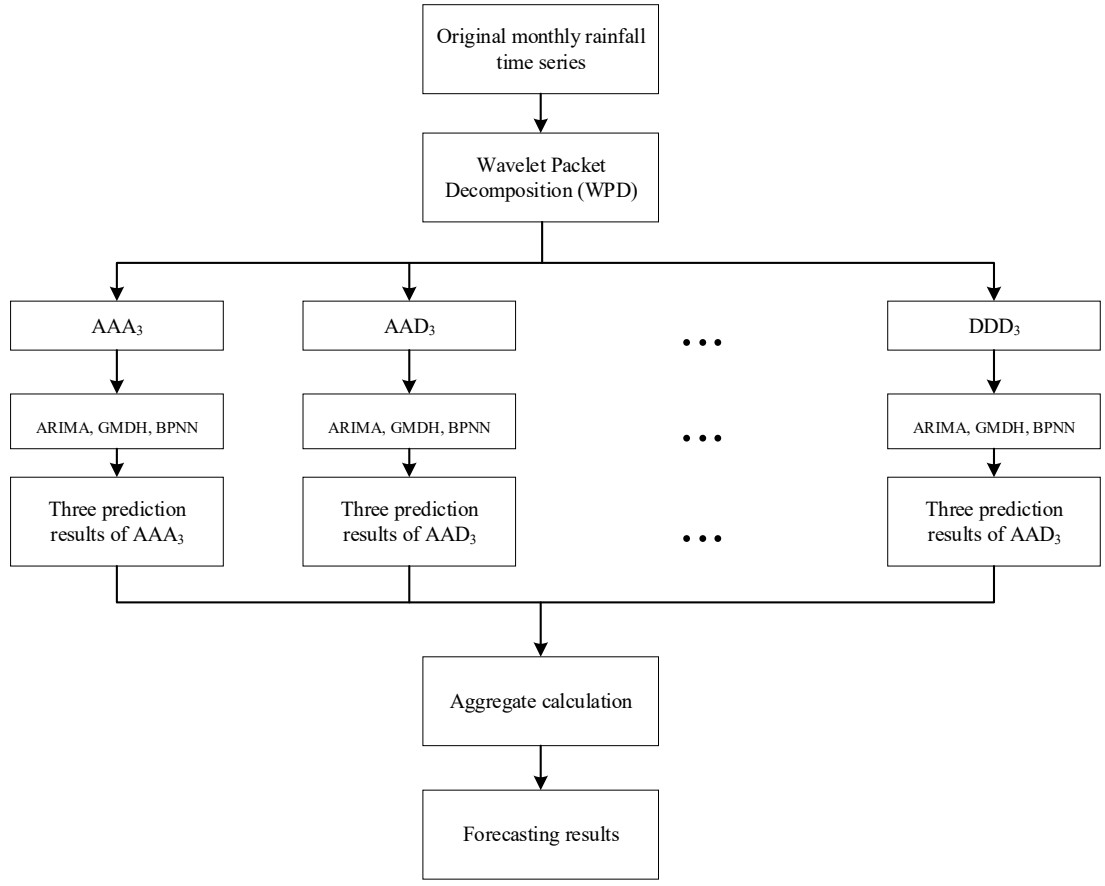

**Figure 7.** The framework of hybrid models.

Step 1: Observed monthly rainfall series are decomposed into eight subsequences with different frequencies and spatiotemporal resolutions, four low-frequency series, and four high-frequency series using WPD.

Step 2: In this study, ACF and PACF are employed to select the number of input variables for the model, and then set values of basic model parameters.

Step 3: ARIMA, BPNN, and GMDH models are used as forecasting tools to model and predict each decomposed sub-sequence separately.

Step 4: Finally, the ensemble monthly rainfall forecasting result of model is formulated by summing the outputs of all submodules.

To sum up, the hybrid WPD-ARIMA, WPD-BPNN, and WPD-GMDH forecasting models use the idea of "decomposition and ensemble". The paper takes 35-year monthly rainfall data from Luoning and Zuoyu stations in Luoyang, China as the test cases.

## 3. Results

### 3.1. Decomposition Results Using WPD and Input Variables Determination

The original monthly rainfall time series are decomposed into eight subsequences with different frequencies and amplitudes using the WPD method. The frequency characteristics of each subsequence are different, and each sub-series plays a different role in the original dataset. The results of WPD of the original monthly rainfall time series data at level 3 are shown in Figures 8 and 9.

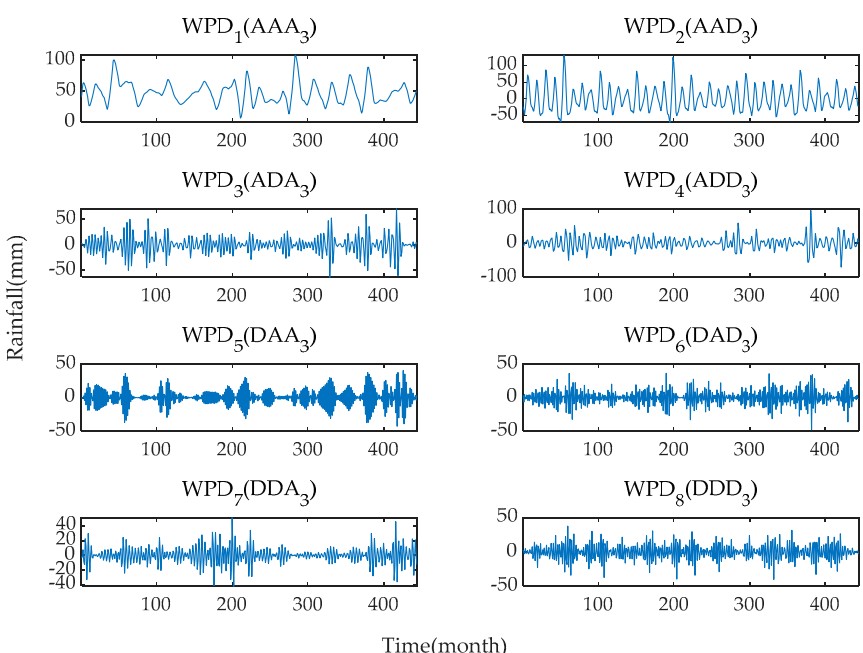

**Figure 8.** Decomposed results for monthly rainfall at Luoning station.

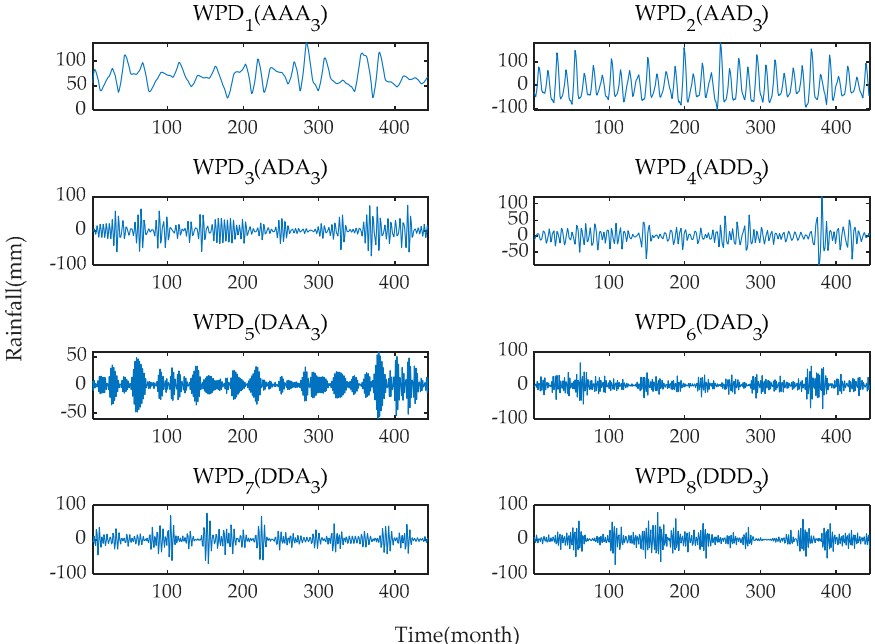

**Figure 9.** Decomposed results for monthly rainfall at Zuoyu station.

Generally, it is very important to set an appropriate number of input variables for data-based prediction models because it is closely related to the characteristics of system to be modeled [62]. In this paper, ACF and PACF are selected as the potential indicators for determining the appropriate input variable. ACF and PACF are normally utilized to pre-determine the sequence of the autoregressive process and modeling of time series [63]. Figures 10 and 11 show ACF and PACF values of the original precipitation series for Luoning and Zuoyu stations, whilst the values of ACF and PACF for all decomposed subseries are not presented here. Referring to ACF and PACF values of the series and influencing factors of precipitation, Table 2 lists input variables of the original series and

their subsequences at Luoning and Zuoyu stations. Among them, $q_{t-p}$ represents the $p^{\text{th}}$ variable before the target output variable.

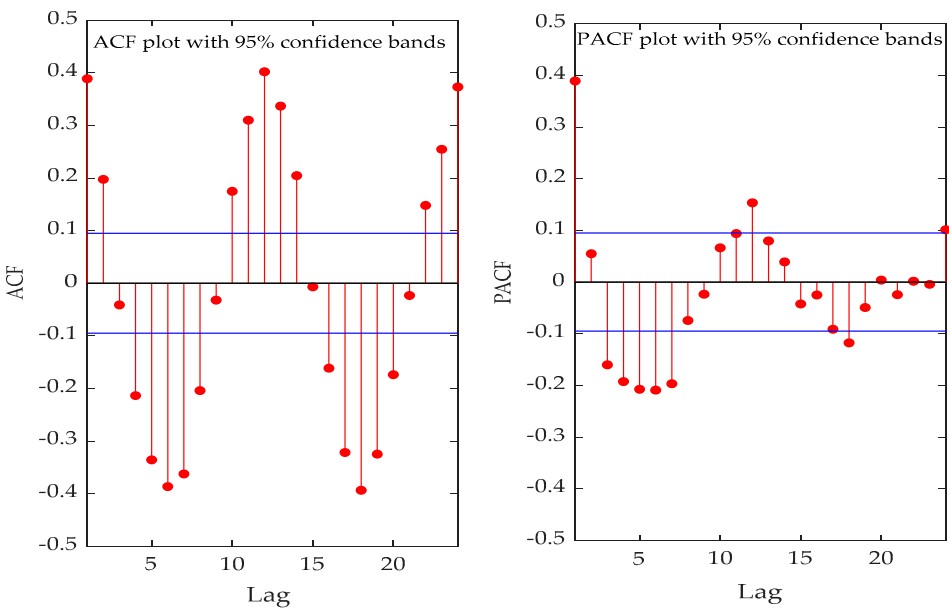

**Figure 10.** ACF and PACF values for the original data series from Luoning station.

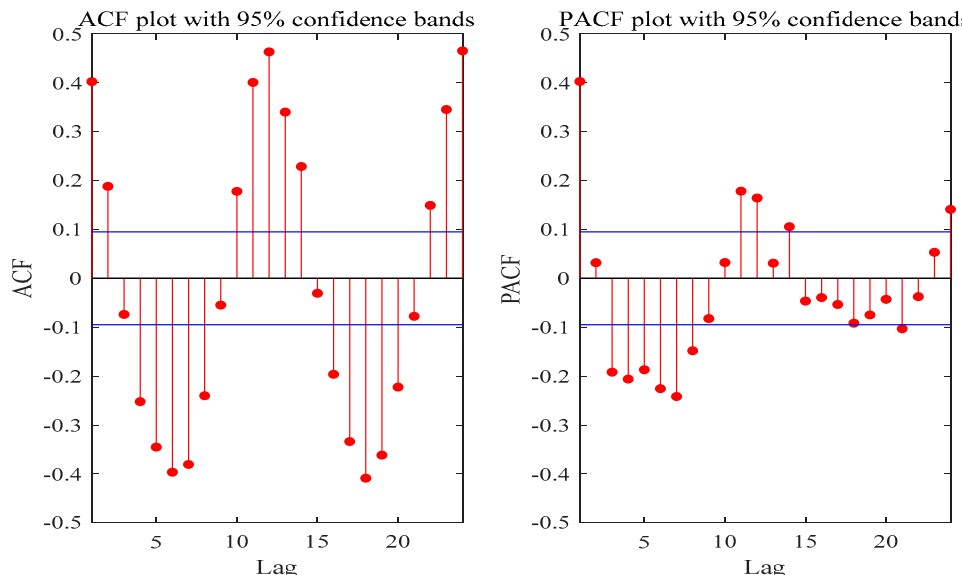

**Figure 11.** ACF and PACF values for the original data series from Zuoyu station.

**Table 2.** Number of input variables for different data series from Luoning and Zuoyu stations based on ACF and PACF analysis.

| No. | Series | Input Variables | |
|---|---|---|---|
| | | **Luoning Station** | **Zuoyu Station** |
| 1 | Original | $x_{t-1} \sim x_{t-12}$ | $x_{t-1} \sim x_{t-13}$ |
| 2 | $WPD_1$ | $x_{t-1} \sim x_{t-12}$ | $x_{t-1} \sim x_{t-12}$ |
| 3 | $WPD_2$ | $x_{t-1} \sim x_{t-12}$ | $x_{t-1} \sim x_{t-12}$ |
| 4 | $WPD_3$ | $x_{t-1} \sim x_{t-12}$ | $x_{t-1} \sim x_{t-13}$ |
| 5 | $WPD_4$ | $x_{t-1} \sim x_{t-11}$ | $x_{t-1} \sim x_{t-11}$ |
| 6 | $WPD_5$ | $x_{t-1} \sim x_{t-12}$ | $x_{t-1} \sim x_{t-12}$ |
| 7 | $WPD_6$ | $x_{t-1} \sim x_{t-12}$ | $x_{t-1} \sim x_{t-12}$ |
| 8 | $WPD_7$ | $x_{t-1} \sim x_{t-13}$ | $x_{t-1} \sim x_{t-13}$ |
| 9 | $WPD_8$ | $x_{t-1} \sim x_{t-11}$ | $x_{t-1} \sim x_{t-13}$ |

*3.2. Model Development*

Six models, namely BPNN, WPD-BPNN, GMDH, WPD-GMDH, ARIMA, and WPD-ARIMA models, are employed for benchmark comparison to study the prediction performance of these conjunction methods.

(1) ARIMA

Generally, the ARIMA model based on the difference process is applied to the modeling of non-stationary series. In this paper, the stationarity of the original monthly rainfall series and subsequences are tested by the Augmented Dickey–Fuller (ADF) test. The results of ADF unit root tests are shown in Table 3. The *h* value of the original and all subsequences of the two stations are zero. The *p*-value of the original sequence and all sub sequences of the two stations is zero, except that the *p*-value of the original sequence of Zuoyu station is 0.0004. When $h = 1$, *p*-value < 0.05, and the value of *t*-statistic is less than the preset upper limit, the null hypothesis is rejected, and the sequence can be considered as stationary; otherwise, the series needs to be differential. It can be seen from Table 3 that the sample set data is stationary series without a single root effect.

**Table 3.** ADF test in the sample data set.

| Name | Sample Data Set | *t*-Statistic Value | Critical Value |
|---|---|---|---|
| Luoning | Original | −5.85207 | −3.42041 |
| | $WPD_1$ | −6.70412 | −3.42041 |
| | $WPD_2$ | −14.33 | −3.42041 |
| | $WPD_3$ | −12.6215 | −3.42041 |
| | $WPD_4$ | −16.2217 | −3.42041 |
| | $WPD_5$ | −9.58776 | −3.42041 |
| | $WPD_6$ | −18.1806 | −3.42041 |
| | $WPD_7$ | −14.919 | −3.42041 |
| | $WPD_8$ | −24.0394 | −3.42041 |
| Zuoyu | Original | −4.86539 | −3.42041 |
| | $WPD_1$ | −6.63558 | −3.42041 |
| | $WPD_2$ | −14.6596 | −3.42041 |
| | $WPD_3$ | −12.2977 | −3.42041 |
| | $WPD_4$ | −16.5301 | −3.42041 |
| | $WPD_5$ | −10.0685 | −3.42041 |
| | $WPD_6$ | −17.6399 | −3.42041 |
| | $WPD_7$ | −14.3801 | −3.42041 |
| | $WPD_8$ | −25.9766 | −3.42041 |

The next step is to choose the optimal ARIMA (*p*, *d*, *q*) model, and the best fitted values of *p* and *q* are selected according to the BIC method. ACF and PACF are used to predetermine the structure of data sets. Furthermore, referring to the BIC minimum criterion, the best fitting model is determined for the original sequence and the decomposed subsequence of the two stations. The values of *p* and *q* are determined based on ACF and

PACF, and the significance test has to be passed, that is, when $p$-value is less than 0.05, select the parameter with minimum BIC statistics. ARIMA models for various sequences are shown in Table 4. The decomposed sub-sequences of Luoning and Zuoyu stations are modeled by the ARIMA model. The original time series are modeled by seasonal ARIMA model (SARIMA), where $p$, $d$, and $q$ represent the autoregressive term, the order of difference, and the moving average term of SARIMA model, respectively.

**Table 4.** The structure of each sequence.

| Name | Sample Data Set | ARIMA ($p, d, q$)/SARIMA ($p, d, q$) ($P, D, Q$) | BIC |
|---|---|---|---|
| Luoning | Original | SARIMA (5,1,1) (1,1,1) | 7.602 |
| | WPD$_1$ | ARIMA (2,1,3) | 1.005 |
| | WPD$_2$ | ARIMA (2,0,8) | 4.016 |
| | WPD$_3$ | ARIMA (2,0,7) | 2.091 |
| | WPD$_4$ | ARIMA (3,0,5) | 3.333 |
| | WPD$_5$ | ARIMA (5,0,7) | −0.274 |
| | WPD$_6$ | ARIMA (2,0,7) | 1.469 |
| | WPD$_7$ | ARIMA (2,0,7) | 1.098 |
| | WPD$_8$ | ARIMA (6,0,8) | 1.132 |
| Zuoyu | Original | SARIMA (5,1,1) (1,1,1) | 8.119 |
| | WPD$_1$ | ARIMA (2,0,5) | 1.469 |
| | WPD$_2$ | ARIMA (3,0,3) | 4.844 |
| | WPD$_3$ | ARIMA (2,0,8) | 2.209 |
| | WPD$_4$ | ARIMA (2,0,8) | 2.992 |
| | WPD$_5$ | ARIMA (10,0,7) | -0.128 |
| | WPD$_6$ | ARIMA (3,0,8) | 1.745 |
| | WPD$_7$ | ARIMA (2,0,7) | 1.43 |
| | WPD$_8$ | ARIMA (6,0,4) | 2.229 |

(2) BPNN

A conventional three-layer BPNN is used to establish the prediction model of monthly precipitation series in this paper. Tan-sigmoid is the transfer function between output and hidden layers, and the nonlinear Levenberg–Marquardt (LM) algorithm is the training function of BPNN. The maximum number of iterations is 100. The number of input layer nodes is the same as the number of input variables. The optimal value is determined by continuously adjusting the number of hidden layer neurons in the range of 2 to 13. The original dataset falls into training samples and test samples. According to the four quantitative indexes, a cross-validation approach is utilized to determine the number of hidden neurons. With the increase of the number of hidden neurons, variations in the statistical indicators of Luoning/Zuoyu station corresponding to different hidden layer nodes are shown in Figures 12 and 13. In this paper, we use $p$ to refer to the number of hidden layers. It can be observed from Figures 12 and 13 that $p$ is not highly correlated with the performance of BPNN model. For Luoning station, when $p = 8$, RMSE and MSE of training and testing periods are both at a minimum, while R and NSEC reach a maximum. For Zuoyu station, when $p = 8$, MSE and RMSE of the testing set reach the minimum value; meanwhile, NSEC and R attain the maximum value. However, when $p$ is seven, MSE and RMSE of the training set reach the minimum value, NSEC and R of the training set reach the maximum value. Therefore, $p$ is chosen to be eight for both Luoning and Zuoyu stations.

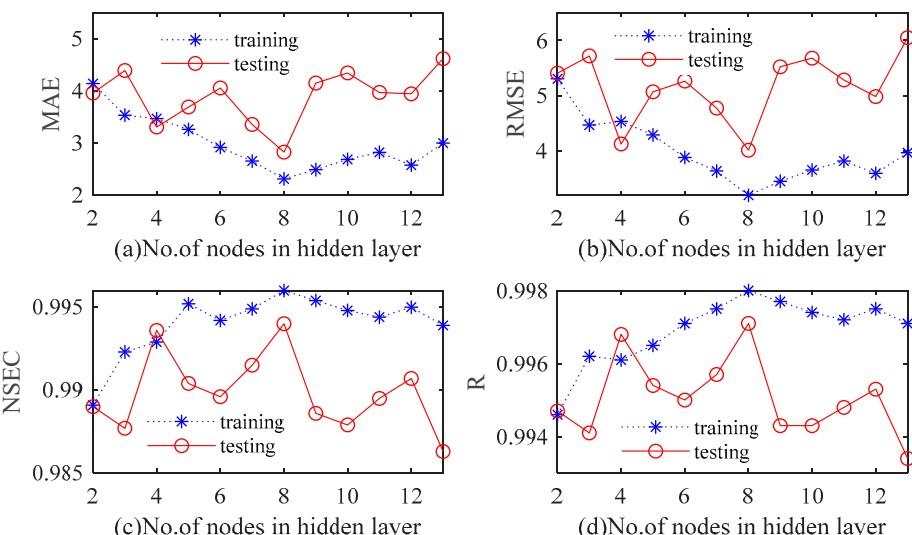

**Figure 12.** Variation of statistical indicators with the number of hidden layer nodes for Luoning station.

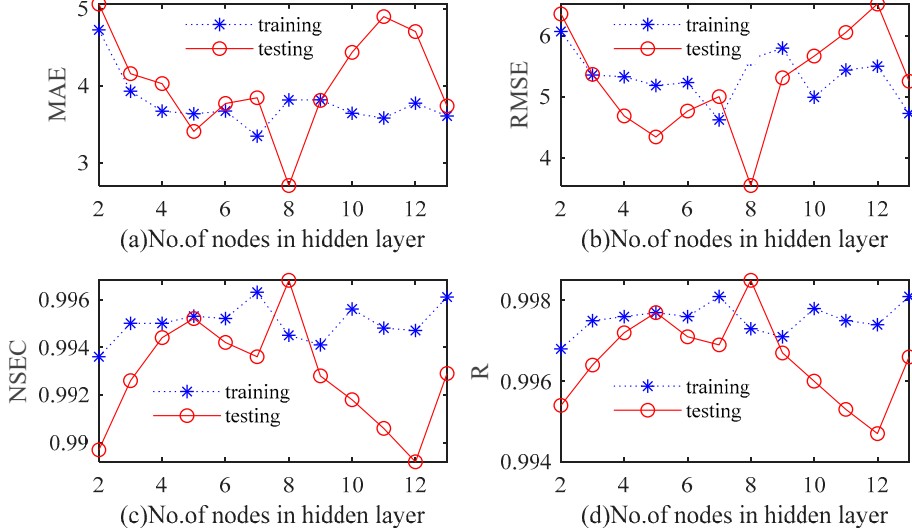

**Figure 13.** Variation of statistical indicators with the number of hidden layer nodes for Zuoyu station.

(3) GMDH

The number of input layer nodes is the same as the number of input variables, and then the regression of output value of upper layer is computed to create the second layer network. GMDH uses the best new variables in each layer to build the next layer network. The GMDH model includes three parameters, namely *a* denoting the maximum number of layers, *b* denoting the maximum number of nodes in each layer, and *p* denoting the selection pressure. In this paper, *a* and *b* are determined as 3 and 15, respectively, whilst *p* is set equal to 0.75 via a trial-and-error method, and the convergence criteria is RMSE. This paper determines an appropriate maximum number of hidden layers and nodes of GMDH model by a trial-and-error method. We set *a* equal to 2, 3, and 5, and *b* equal to 5, 10, and 15. The results (not supplied) show that the numbers of *a* and *b* have a significant effect on the performance of the GMDH model.

(4) WPD

WPD is adopted for data preprocessing, which can eliminate noises in hydrological time series. The selection of an appropriate mother wavelet is very significant to WPD. The Symlet wavelet function is an improved version of the classical Daubechies wavelet function, which evades the change of waveform in the process of signal decomposition [64].

Therefore, the fourth order Symlet wavelet function is considered as the mother wavelet function. In this paper, three-scale wavelet WPD is selected because large-scale wavelet packet decomposition may lead to information loss.

### 3.3. Results and Discussion

Based on the above description, different methods are utilized to model the observed rainfall and extracted sub-sequences. Tables 5 and 6 list the statistical indexes of different algorithms for Luoning and Zuoyu stations during training and testing periods.

**Table 5.** Forecasting performance indices of models for Luoning station.

| Model | Training | | | | Testing | | | |
|---|---|---|---|---|---|---|---|---|
| | R | RMSE | MAE | NSEC | R | RMSE | MAE | NSEC |
| ARIMA | 0.608 | 40.926 | 27.442 | 0.352 | 0.459 | 46.474 | 27.792 | 0.191 |
| WPD-ARIMA | 0.984 | 9.210 | 7.298 | 0.967 | 0.988 | 8.224 | 6.060 | 0.975 |
| BPNN | 0.667 | 37.896 | 25.913 | 0.445 | 0.484 | 45.775 | 29.204 | 0.215 |
| WPD-BPNN | 0.998 | 3.296 | 2.384 | 0.996 | 0.997 | 4.054 | 2.912 | 0.994 |
| GMDH | 0.584 | 41.299 | 28.495 | 0.340 | 0.600 | 41.844 | 24.575 | 0.344 |
| WPD-GMDH | 0.970 | 12.372 | 9.588 | 0.941 | 0.966 | 13.734 | 11.171 | 0.929 |

**Table 6.** Forecasting performance indices of models for Zuoyu station.

| Model | Training | | | | Testing | | | |
|---|---|---|---|---|---|---|---|---|
| | R | RMSE | MAE | NSEC | R | RMSE | MAE | NSEC |
| ARIMA | 0.679 | 55.717 | 38.124 | 0.459 | 0.576 | 53.689 | 31.856 | 0.263 |
| WPD-ARIMA | 0.987 | 12.455 | 9.525 | 0.973 | 0.992 | 7.970 | 6.415 | 0.984 |
| BPNN | 0.709 | 53.518 | 36.089 | 0.500 | 0.607 | 50.37 | 31.846 | 0.352 |
| WPD-BPNN | 0.997 | 5.935 | 4.102 | 0.994 | 0.998 | 3.705 | 2.889 | 0.996 |
| GMDH | 0.662 | 56.751 | 39.460 | 0.433 | 0.643 | 48.271 | 30.439 | 0.405 |
| WPD-GMDH | 0.973 | 17.771 | 13.970 | 0.945 | 0.980 | 14.797 | 11.623 | 0.944 |

For Luoning station, the WPD-BPNN model attains the best RMSE, MAE, R, and NSEC values during the training period, which are 3.292, 2.384, 0.998, and 0.956, respectively. In the testing phase, the WPD-BPNN model also attains the best R, RMSE, MAE, and NSEC statistics of 0.997, 4.054, 2.912, and 0.994, respectively. Meanwhile, for Zuoyu station, the WPD-BPNN model attains the best RMSE, MAE, R, and NSEC values during the training period, which are 5.935, 4.102, 0.997, and 0.994, respectively. In analyzing the results during the testing phase, the WPD-BPNN model attains the best R, RMSE, MAE, and NSEC statistics of 0.998, 3.705, 2.889, and 0.996, respectively. Referring to the four evaluation indicators in this paper, WPD-BPNN can attain the best performance in monthly precipitation prediction.

Tables 7 and 8 list the comparison of results on model prediction performance by different indicators. When forecasting monthly rainfall at Luoning station, WPD-BPNN is able to attain the best improving capability of RMSE and MAE in the training phase, while WPD-GMDH is able to attain the best improving capability of R and NSEC in the training phase. In analyzing the figures during the testing phase, WPD-BPNN attains the best improving capability of RMSE and MAE, while WPD-ARIMA attains the best improving capability of R and NSEC. In addition, it can be seen from Table 8 that the prediction performance of the models is similar for Zuoyu and Luoning stations. Therefore, the monthly rainfall series decomposed by WPD method as the input of BPNN model can drastically improve the forecasting accuracy. This reaffirms the superior performance

of WPD. Furthermore, the enhancement capabilities of different evaluation methods are different in terms of different phases and different forecasting measures.

**Table 7.** Comparison of results of model prediction performance for Luoning station.

| Model | Index | Training (%) | Testing (%) |
|---|---|---|---|
| WPD-ARIMA & ARIMA | R($\uparrow$) | 61.81 | 115.48 |
| | NSEC($\uparrow$) | 174.54 | 732.71 |
| | RMSE($\downarrow$) | 77.5 | 80.3 |
| | MAE($\downarrow$) | 73.4 | 78.19 |
| WPD-BPNN & BPNN | R($\uparrow$) | 45.52 | 106.8 |
| | NSEC($\uparrow$) | 123.98 | 362.45 |
| | RMSE($\downarrow$) | 91.3 | 91.14 |
| | MAE($\downarrow$) | 90.8 | 90.3 |
| WPD-GMDH & GMDH | R($\uparrow$) | 66.22 | 61.17 |
| | NSEC($\uparrow$) | 176.38 | 170.22 |
| | RMSE($\downarrow$) | 70.04 | 74.35 |
| | MAE($\downarrow$) | 66.35 | 54.54 |

Note: ($\uparrow$) represents the percentage of performance improvement of the new model compared to the original model, and ($\downarrow$) represents the percentage of performance reduction of the new model compared to the original model.

**Table 8.** Comparison of results of model prediction performance for Zuoyu station.

| Model | Index | Training (%) | Testing (%) |
|---|---|---|---|
| WPD-ARIMA&ARIMA | R($\uparrow$) | 45.34 | 72.08 |
| | NSEC($\uparrow$) | 105.30 | 273.50 |
| | RMSE($\downarrow$) | 77.65 | 85.15 |
| | MAE($\downarrow$) | 75.02 | 79.86 |
| WPD-BPNN&BPNN | R($\uparrow$) | 40.53 | 64.51 |
| | NSEC($\uparrow$) | 98.86 | 183.50 |
| | RMSE($\downarrow$) | 88.91 | 92.64 |
| | MAE($\downarrow$) | 88.63 | 90.93 |
| WPD-GMDH&GMDH | R($\uparrow$) | 46.81 | 52.33 |
| | NSEC($\uparrow$) | 118.32 | 133.37 |
| | RMSE($\downarrow$) | 68.69 | 69.35 |
| | MAE($\downarrow$) | 64.60 | 61.82 |

Note: where ($\uparrow$) represents the percentage of performance improvement of the new model compared to the original model, and ($\downarrow$) represents the percentage of performance reduction of the new model compared to the original model.

For the two research objects in this paper, the performance of all models during training and test periods are shown in Figures 14–17. The performances of hybrid models for monthly rainfall simulation are able to attain better performance than those of conventional ARIMA, BPNN, and GMDH methods. WPD-BPNN presents the best performance, and its trend line is almost perfectly close to the smooth line of the observed data. In contrast, there are huge deviations between the prediction results obtained by ARIMA, BPNN, and GMDH methods and observed data. In addition, the prediction values of the extreme points of the three single models are far less than the observed value, and the peak prediction also has an obvious lag effect. However, compared with ARIMA, GMDH, and BPNN, the three WPD-based models have greatly improved the peak value accuracy and time positioning. Meanwhile, the models prior to improvement cannot capture abrupt changes of precipitation in rainy season. Therefore, compared with several existing methods in this paper, WPD-BPNN is the most efficient tool for monthly rainfall forecasting, since it can achieve excellent prediction results.

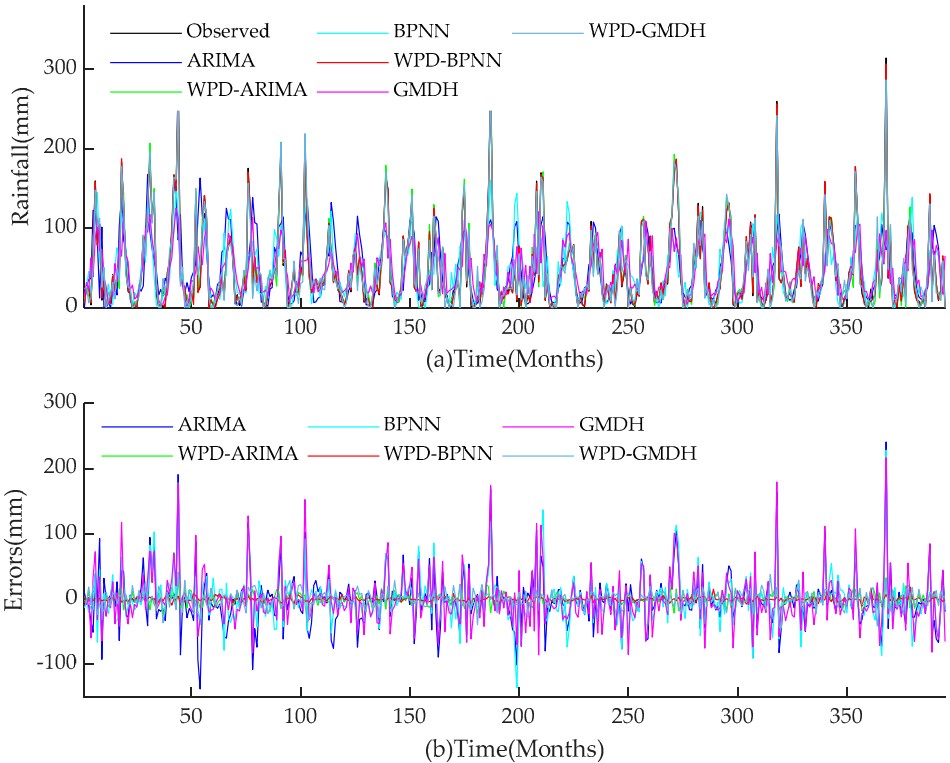

**Figure 14.** (**a**) Forecasting results of Luoning station in the training period; (**b**) forecasting errors of Luoning station in the training period.

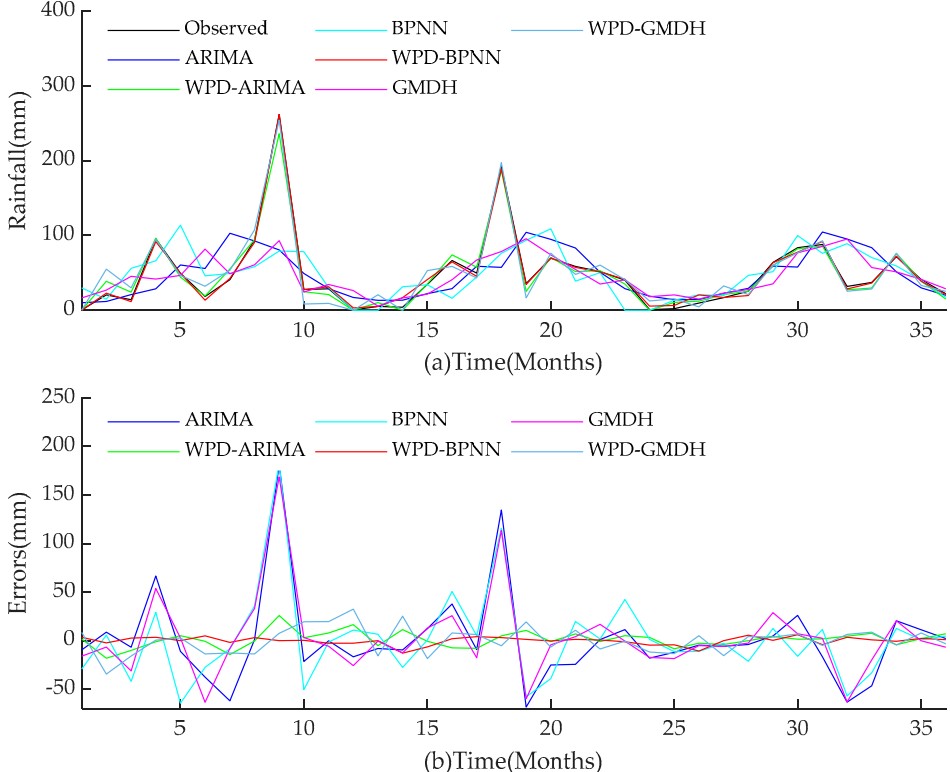

**Figure 15.** (**a**) Forecasting results of Luoning station in the testing period; (**b**) forecasting errors of Luoning station in the testing period.

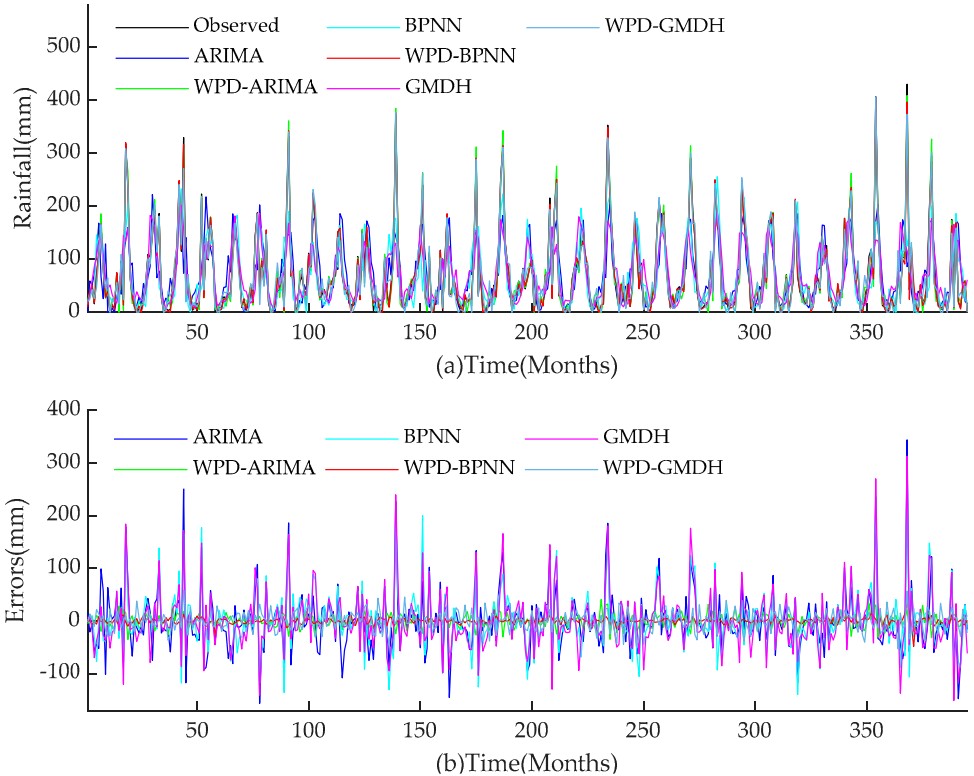

**Figure 16.** (**a**) Forecasting results of Zuoyu station in training period; (**b**) forecasting errors of Zuoyu station in the training period.

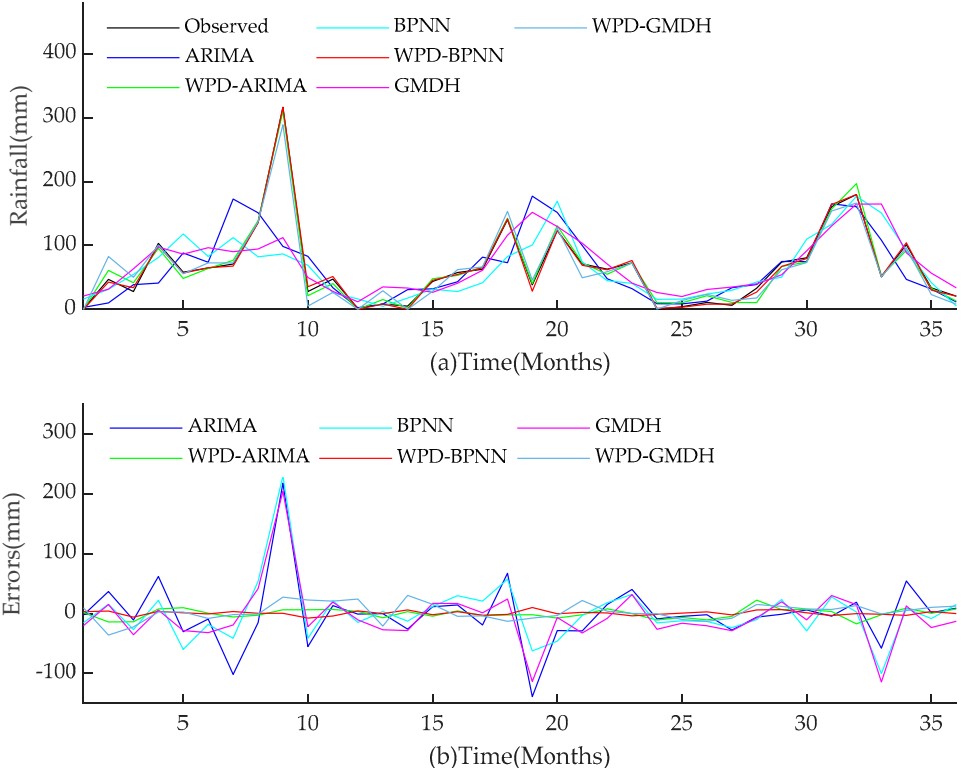

**Figure 17.** (**a**) Forecasting results of Zuoyu station in testing period; (**b**) forecasting errors of Luoning station in the testing period.

## 4. Conclusions

In recent years, the improvement of hydrological forecasting accuracy has attracted widespread attention around the world. In order to broaden the scope of hydrological forecasting theory, this study explores the performance of several data-driven methods based on WPD in monthly precipitation forecasting. Firstly, the observed monthly rainfall time series are decomposed into eight subsequences with different frequencies and spatiotemporal resolutions by WPD. Then, three data-based models, namely BPNN, GMDH, and ARIMA models, are utilized to complete the prediction for the decomposed monthly rainfall series, respectively. Finally, the ensembled prediction result of the model is formulated by summing the outputs of all submodules. Monthly rainfall data from two stations in China are utilized to test the performance of these methods. To evaluate the forecast capacity of different models, four standard statistical metrics are adopted to estimate the global and local errors of the models.

The results reveal that the WPD model is suitable for the decomposition of monthly rainfall series, and WPD-BPNN can provide the best performance during both training and testing periods in terms of the four evaluation indicators in this paper. The following briefly introduces the advantages of the WPD-BPNN method. Firstly, the principle of WPD is simple and inclusive, and it can comprehensively and deeply analyze the characteristics of monthly precipitation series. Secondly, the prediction performance of BPNN only depends on the characteristics of input variables. Finally, the proposed model does not require complex decision-making for the explicit form of the model in different cases. Therefore, the hybrid forecast model based on WPD technology is an efficient tool to improve the accuracy of mid- and long-term rainfall forecasting.

It should be pointed out that, although this paper has fully verified the feasibility of WPD-BPNN in monthly precipitation forecasting, there are still several limitations to be explored in the future research. Firstly, the study is carried out based on two time series, so we will test the generalization of the proposed model. The second is to test the performance of other algorithms combined with WPD. The last major issue is to develop an appropriate optimization algorithm to improve the performance of WPD-BPNN. In future research, it is necessary to conduct in-depth research on the three aspects above to explore more efficient and accurate forecasting techniques and make contributions to the field of hydrological forecasting.

**Author Contributions:** W.W.: Conceptualization, Methodology, Writing—original draft. Y.D.: Program implementation, data curation, Writing—original draft preparation. K.C.: Writing and editing—original draft. H.C.: Writing—original draft. C.L.: Investigation. Q.M.: Formal analysis. All authors have read and agreed to the published version of the manuscript.

**Funding:** The project of Key Science and Technology of the Henan province (202102310259; 202102310588), and the Henan province University Scientific and Technological Innovation team (No: 18IRTSTHN009).

**Institutional Review Board Statement:** Not applicable.

**Informed Consent Statement:** Not applicable.

**Data Availability Statement:** All authors made sure that all data and materials support our published claims and comply with field standards.

**Conflicts of Interest:** The authors declare that they have no conflict of interest.

## Abbreviations

| | |
|---|---|
| WPD | wavelet packet decomposition |
| BPNN | back-propagation neural network |
| GMDH | group method of data handing |
| ARIMA | autoregressive integrated moving average |
| ANN | artificial neural network |
| GP | genetic programming |
| SVM | support vector machines |
| ANFIS | adaptive neuro-fuzzy inference system |
| AR | auto-regressive |
| MA | moving average |
| ARMA | autoregressive moving average |
| LM | Levenberg–Marquardt |
| EMD | empirical mode decomposition |
| EEMD | ensemble empirical mode decomposition |
| FT | Fourier transform |
| SVR | support vector regression |
| QPSO | quantum-behaved particle swarm optimization |
| VMD | variational mode decomposition |
| LSWA | least-squares wavelet analysis |
| WD | wavelet decomposition |
| DWT | discrete wavelet transform |
| WR | wavelet representation |
| LPF | low-pass filter |
| HPF | high-pass filter |
| RMSE | root mean-squared error |
| MAE | mean absolute error |
| R | coefficient of correlation |
| NSEC | Nash–Sutcliffe efficiency coefficient |
| ACF | autocorrelation function |
| PACF | partial autocorrelation function |
| ADF | augmented Dickey–Fuller |
| BIC | Bayes information criteria |
| SCS-CN | soil conservation service-curve number |

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
