# Peer review of "A Comparison of BPNN, GMDH, and ARIMA for Monthly Rainfall Forecasting Based on Wavelet Packet Decomposition"

_water, doi:10.3390/w13202871_

Round 1

Reviewer 1 Report

The title of this paper is “A comparison of BPNN, GMDH and ARIMA for monthly rainfall forecasting based on Wavelet Packet Decomposition”. The title of this paper is interesting. However, this paper needs a revision for publication in Water. Therefore, I recommend a major revision in this paper.

Author Response

Reply to questions or reviews given by Reviewer #1:

The title of this paper is “A comparison of BPNN, GMDH and ARIMA for monthly rainfall forecasting based on Wavelet Packet Decomposition”. The title of this paper is interesting. However, this paper needs a revision for publication in Water. Therefore, I recommend a major revision in this paper.

Comment: Keywords should be written in uppercase or lowercase letters according to the form.

Response: Thank you very much for your suggestion. We amend the form of ‘Keyword’ in the revised manuscript.

Comment: In introduction, the limitations of former studies should be mentioned and the merits of this study to overcome them should be explained.

Response: Thank you very much for the comment. We add the limitations of former studies and the merits of this study in Section 1 introduction.

Comment: An explanation of the parameters used in all equations is required.

Response: Thank you very much for the suggestion. We check symbols of all equations in the manuscript and add explanation to the missing ones.

Comment: Figures 1 and 2 needs an explanation.

Response: Thank you very much for this suggestion. We modify the symbols in Figures 1 and 2 to align with the corresponding equations in the paper, and the detailed explanation is shown under the equations.

Comment: In Section 4 (Case study), a watershed map is required to describe the target watershed.

Response: Thank you very much for this suggestion. We add a watershed map to describe the target watershed in Section 2.1 Study region.

Comment: In Figures 7 and 8, the numerical value for the 95% confidence bands should be written on the graphs.

Response: Thank you very much for this suggestion. We revise Figures 7 and 8 based on the comment, and the revised Figures are in Section 3.1 Decomposition results using WPD and input variables determination Figures 10 and 11.

Comment: In Table 2, why are the input variables different for each series.

Response: Thank you very much for this suggestion. We use WPD to decompose an original monthly rainfall series into a sequence of subseries with different frequencies and spatiotemporal resolutions. In order to ensure the optimal prediction accuracy of each subsequence, the input variables of different sequences are different.

Comment: In Table 3, why are all critical values the same.

Response: Thank you very much for this suggestion. For ADF test, p-Value <0.05, 5% test critical values is -3.420406.

Comment: In conclusion, it is required to mention the limitations of this study and follow-up studies to overcome them.

Response: Thank you very much for this suggestion. We add the limitations and future work of the study in Section 4 Conclusion.

The above responses are our reply to the reviews about water-1389911. We look forward to hearing further information from you.

Sincerely yours

Reviewer 2 Report

The comments for the authors are to be found in the uploaded document.

Author Response

Reply to questions or reviews given by Reviewer #2:

Introduction

Comment: In the introduction, the authors provided a very critical and pertinent review of the types of runoff prediction models. Thus, they justified very well the reasons for choosing one or another of the models, to the detriment of others, depending on their capabilities. Despite the consistency of the well explained models – related part of the introduction, the lines 105-107 are somewhat unclear. I suggest to better explain the two steps followed and what that separation on data-based models consists of, as well as their combination at the end.

Also, in line 108, I would avoid using the term “the remainder” and I would just leave “the paper is arranged as follows”, because otherwise you leave the impression that what follows is only the remaining small part of the article and that the rest has already been explained before.

Response: Thank you very much for this comment. We realize that the description of forecasting steps of models is inaccurate, so we modify part of Section 1 Introduction based on this comment.

Methodology

Comment: In section 2 (lines 112-173), where the three categories of models are presented, references to the types of data to which these models can be applied in this case are missing. The models are presented only in general, as we can find in many papers, some of which are cited right here, but are not discussed in full understanding of their applicability in the context of this study. Therefore, I suggest you exemplify in a few lines, in each of the three models, to which element correspond the parameters employed in the formulas, using the reasoning based on the precipitation data that are the subject of this article. Otherwise, these descriptions of the models may remain disconnected from the specific understanding of the types of data and parameters (from the monthly rainfall databases used) on which they are tested.

Response: Thank you very much for this comment. Based on this comment, we add the elements corresponding to the parameters used in the formula in Section 2.3 Methods.

Comment: In sub-chapter 2.5 The Hybrid Forecasting Model, line 219, “different frequencies and spatiotemporal resolutions” are mentioned. Please, be more specific what you mean by frequencies and how you define the spatial resolution, if dealing with data from two stations.

Response: Thank you very much for this comment. From the perspective of function theory, WPD projects the signal into the space expanded by wavelet packet basis function. From the perspective of signal preprocessing, it allows the signal to pass through a series of filters with different center frequencies but the same bandwidth. A function in the wavelet library includes scale coordinates, position coordinates and oscillation times. The input signal is decomposed into a series of subsequences with different amplitudes and frequencies by wavelet function, that is, its temporal and spatial distributions are changed.

Comment: In lines 228-229, the sentence “use the idea of decomposition and ensemble” is not clear. Should we understand that this is the common feature of the three models.

Response: Thank you very much for this comment. The prediction results of each hybrid model are obtained by linearly accumulating the outputs of each submodule.

Methodology

Comment: In lines 235-248, the models for estimating the errors are presented without referencing the authors. This chapter is inconsistent. I personally do not see it alone, but rather coupled with the other methodological sub-chapters.

Response: Thank you very much for this comment. We add the following citations and organize the paper according to the comment in the revised manuscript.

  1. Gentilucci, M.; Materazzi, M.; Pambianchi, G.; Burt, P.; Guerriero, G. Assessment of Variations in the Temperature-Rainfall Trend in the Province of Macerata (Central Italy), Comparing the Last Three Climatological Standard Normals (1961–1990; 1971–2000; 1981–2010) for Biosustainability Studies. Environmental Processes 2019, 6, 391-412, doi:10.1007/s40710-019-00369-8.
  2. Wang, W.; Lu, Y. Analysis of the Mean Absolute Error (MAE) and the Root Mean Square Error (RMSE) in Assessing Rounding Model. IOP Conference Series: Materials Science and Engineering 2018, 324, 012049, doi:10.1088/1757-899x/324/1/012049.
  3. Kim, H.I.; Keum, H.J.; Han, K.Y. Real-Time Urban Inundation Prediction Combining Hydraulic and Probabilistic Methods. Water 2019, 11, doi:10.3390/w11020293.

Case study

Comment: In lines 252 – 255, it is confusing if Yiluo is the common river basin of Yi and Luo rivers or if there is another river called Yiluo (line 252). Please refer to it as a basin, solely, or as a river, if that’s the case. Also, in the lines 255-256, if you prefer to use the Chinese version of the two names (with “he” at the end), Lyohe and Yihe, please be consistent throughout the paper. If you call them Yi and Luo rivers only (without the suffix “he”), keep it like that, to avoid confusions, as if they were two different rivers. If you find more appropriate, please consider including a map with the river basin, the rivers concerned and the location of the two stations within the watershed under study.

Response: Thank you very much for this comment. We amend all inappropriate expressions in the paper based on this comment. We add a watershed map to describe the target watershed in Section 2.1 Study region

Comment: Line 259 – remove one “m” from “115.6 mmm”.

Response: Thank you very much for this comment. We revise Section 2.1 Study region based on this comment.

Comment: Lines 261 – 262: what do you mean by “several conjunction methods”?

Response: Thank you very much for this comment. We replace " several conjunction methods " with " several prediction methods ".

Comment: Lines 273 – 274 are very ambiguous. How the data series were decomposed and what the “spatiotemporal” resolution means for some monthly data at two stations?

Response: Thank you very much for this comment. We replace " with different frequencies and spatiotemporal resolutions" with " with different frequencies and amplitudes ". From the perspective of function theory, WPD projects the signal into the space expanded by wavelet packet basis function. From the perspective of signal preprocessing, it allows the signal to pass through a series of filters with different center frequencies but the same bandwidth. A function in the wavelet library includes scale coordinates, position coordinates and oscillation times. The input signal is decomposed into a series of subsequences with different amplitudes and frequencies by wavelet function, that is, its temporal and spatial distributions are changed.

Comment: Line 275: what role does each sub-series play? It is unclear what is meant by this statement.

Response: Thank you very much for this comment. The sum of the eight subsequences obtained by WPD is equal to the original time series.

Comment: Line 276 – what does level 3 imply?

Response: Thank you very much for this comment. Four subsequences are obtained by two-layer WPD and eight subsequences are obtained by three-layer WPD. In this paper, a three-layer WPD method is used to preprocess the original rainfall time series.

Comment: In lines 277 – 279, the phrase lacks subject and meaningfulness. Please consider revising it!

Response: Thank you very much for this comment. We revise the sentence as follows:

Generally, it is very important to set the appropriate number of input variables for data-based prediction models, because it is closely related to the characteristics of system to be modeled.

Comment: Line 279: which are the important parameters? Line 279: which are the important parameters?

Response: Thank you very much for this comment. We revise the sentence as follows:

In this paper, ACF and PACF are selected as potential indicators for determining appropriate input variables.

Comment: In lines 287 – 291, which are the units of the parameters plotted in the graphs? Also, a general question would be how the dataset was decomposed, based on which criteria and with what kind of expected result?

Response: Thank you very much for this comment. We add the units for revised Figures 8 and 9. The structure of WPD are:

T = wpdec(X, N,'wname', E, P) returns a WP tree object T, corresponding to a wavelet packet decomposition of the vector X, at level N, with a particular wavelet.

where E is a character vector containing the type of entropy: E = 'threshold'

P is an optional parameter: P is the threshold (0 <= P).

Comment: Lines 317 – 324: The explanations are too brief. It is not understood what was achieved in each step of the analysis, what criteria were applied and what parameters were chosen.

Response: Thank you very much for this comment. The values of p and q are determined based on ACF and PACF, and the significance test must be passed, that is, when p-value is less than 0.05, select the parameter with minimum BIC statistics.

Comment: Lines 321 – 322: which are those decomposed sub-sequences? How are they organized according to the precipitation values included in each group (chronologically, in ascendent order)? For the Tables 3 and 4, one feels the lack of an explanation of what is comprised in WPD1…8

Response: Thank you very much for this comment. The sub-sequences are WPD1, WPD2, … WPD8, respectively, and they are the third level decomposition low frequency bands 1,2,3,4 and the third level decomposition high frequency bands 5,6,7,8 respectively.

Comment: Line 366: “The selection of the appropriate mother wavelet is very significant to WPD” – unclear. What do you mean by “appropriate mother wavelet” and “significant” attribute?

Response: Thank you very much for this comment. It is important to determine an appropriate mother wavelet. The Symlet wavelet function is an improved version of the classical Daubechies wavelet function, which evades the change of waveform in the process of signal decomposition. Therefore, the fourth order Symlet wavelet function is considered as the mother wavelet function. Available wavelet names are:

Daubechies: 'db1' or 'haar', 'db2', ... ,'db45'

    Coiflets  : 'coif1', ... ,  'coif5'

      Fejer-Korovkin: 'fk4', 'fk6', 'fk8', 'fk14', 'fk18', 'fk22'

    Symlets   : 'sym2' , ... ,  'sym8', ... ,'sym45'

    Discrete Meyer wavelet: 'dmey'

    Biorthogonal:

        'bior1.1', 'bior1.3' , 'bior1.5'

        'bior2.2', 'bior2.4' , 'bior2.6', 'bior2.8'

        'bior3.1', 'bior3.3' , 'bior3.5', 'bior3.7'

        'bior3.9', 'bior4.4' , 'bior5.5', 'bior6.8'.

    Reverse Biorthogonal:

        'rbio1.1', 'rbio1.3' , 'rbio1.5'

        'rbio2.2', 'rbio2.4' , 'rbio2.6', 'rbio2.8'

        'rbio3.1', 'rbio3.3' , 'rbio3.5', 'rbio3.7'

        'rbio3.9', 'rbio4.4' , 'rbio5.5', 'rbio6.8'.

Comment: Line 375: “we can conclude”. It should not be already a conclusion. Before that, discussions should be made around the results.

Response: Thank you very much for this comment. The description of this sentence is inappropriate. This sentence and other inappropriate expressions in Section 3.3 Results and discussion are revised.

Comment: Lines 372 – 384 (general remark): The description of the methods is quite cumbersome and very technical, not offering enough details about the expected results through this complex approach using the cited models. In this discussion part at least, the ideas should be more explicit about the performance of each model. Expressions like “we can conclude”, “in conclusion” are offered several times in an intermediary part (Results and Discussion), where one should not directly jump to the conclusions, but present the outcomes in a comprehensive way. On the other hand, the results for each model are treated very fugitive, by only enumerating some statistics obtained and then quickly drawing the conclusion that WPD-BPNN model is the best. Moreover, in line 279, you state that “Therefore, we can conclude that WPD-BPNN model performs best among all models” (as it were in general, for all the data), while later, in lines 383-384, you conclude that the same WPD-BPNN model is the best for all data. In line 279, isn’t it just referring to Luoning Station?

Response: Thank you very much for this comment. We amend Section 3.3 Results and discussion based on the comments of reviewers #2 and reviewers #3. Please refer to the revised manuscript for details.

Comment: Figures 435-437: the chart is too loaded. It is difficult to visually distinguish which of the models is more efficient. I would suggest that this figure be organized into 6 small figures, each with the observed data and a result of each model. Same remark and suggestion for Figure 15.

Response: Thank you very much for this comment. We improve the quality of Figures 13,14,15,16,17 based on the comment of reviewers #2 and reviewers #3.

Conclusion

Comment: One last question would be related to the testing period of only 3 years. Would it be enough or too small, though? I would suggest that this aspect be discussed at the end of the results and discussions, or even at the conclusions, as a limitation of the study or as a starting point for its continuation in the future, using a longer testing period.

Response: Thank you very much for this comment. We add the limitations and future work of the study in Section 4 Conclusion.

The above responses are our reply to the reviews about water-1389911. We look forward to hearing further information from you.

Sincerely yours

Reviewer 3 Report

Reviewer’s Report on the manuscript entitled:

A comparison of BPNN, GMDH, and ARIMA for monthly rainfall forecasting based on Wavelet Packet Decomposition

The authors investigated the performance of several known methods based on Wavelet Packet Decomposition (WPD) in monthly precipitation forecasting. They utilized three known methods, namely, BPNN, GMDH, and ARIMA to complete the prediction for the decomposed monthly rainfall series. They compared the methods using the data sets from two stations in China and evaluated the forecast capacity of the models.

The manuscript is generally interesting, but it is very disorganized. Below, please find my comments. Please note that the line numbers are not inserted which makes it difficult for the reviewer to provide detailed comments. Please insert the line numbers when you submit the revised version.

Please note that when you prepare the revised manuscript have all the results (including tables and figures) in the result section. Tables and figures for pre-processing of the data can go to the data set and pre-processing section. Thus, I suggest the authors organize the paper as follows and insert the line numbers:

1. Introduction.

2. Materials and Methods.

    2.1 Study region (please include a map of the study region here).

    2.2 Data sets and pre-processing

    2.3 Methods

    2.4 Evaluation indices

3. Results

4. Discussion

5. Conclusion

The data sets analyzed in this manuscript are only two time series! Your current conclusion section must be improved by adding the limitations of the study and future work.

In the first paragraph of the Introduction please also include the following articles. The first one shows the application of a wavelet analysis software package (namely, the least-squares wavelet: LSWAVE) for streamflow and climate analysis and forecasting, and the second article talks about several AI techniques for climate forecasting:

https://doi.org/10.1016/j.ejrh.2021.100847

https://doi.org/10.3390/rs13163209

The least-squares wavelet analysis (LSWA) is a new method of analyzing non-stationary time series that may not be sampled at equally spaced time intervals and can consider observational uncertainties. LSWA has shown promise in a successful analysis of streamflow and climate time series that is a tool in the LSWAVE software (the first reference above). This can also be mentioned in the first paragraph on page 3 toward the end of the Introduction.  

Equations (1) and (2) the variable “x” has a different format in Equation (2) and the first term on the right-hand side of Equation (1). Please check. It is not clear whether Equation (3) is derived from Equations (1) and (2). That should be clarified.

Equation (4) also uses “y” with two different formats. Please check the equations and correct them. The gradient symbol must also be defined.  

Figure 1. The font size of all the variables should be the same and not too large.

Equation (5). Please also write “i = 0, 1, …, n” and remove i(0,1) in the last line of page 4.

Grammar. After Equation (5). It should be “formulas above” not “above formula”.

The three lines below Equation (21): Figure 3 not Figure. 3. Also please never start a sentence with a variable or number. For example, instead of saying “x is the original signal”, please say “The original signal is shown by x”, etc.

What about other symbols (names) in Figure 3? Please clearly describe them in the caption of Figure 3.

I do not see anywhere in the text that the authors refer to Table 1. Please note that all the figures and tables must be clearly referred to in the body of the manuscript.

In the caption of Figures 5 and 6, please mention months since 1980. Is that true? Is it since 1980?

Figures 13,14,15,16. Each figure should have two panels, the top panels are the current figures, but the bottom panels should be the residual time series (the result of each method subtracted from the observed series). This way the readers can easily see the differences. Also, I suggest display them as continuous lines rather than using markers. There are currently so many markers (dots) in Figures 13 and 15, making it very difficult to inspect the figures.

Abbreviations. All acronyms in the manuscript should be defined the first time they appear. The full name should come first followed by its abbreviation inside parentheses. Please be consistent. Also, please add an acronym table at the end of the manuscript listing all the acronyms used in the manuscript.  

Thank you for your contribution

Regards,

Author Response

Reply to questions or reviews given by Reviewer #3:

Comment: The manuscript is generally interesting, but it is very disorganized. Below, please find my comments. Please note that the line numbers are not inserted which makes it difficult for the reviewer to provide detailed comments. Please insert the line numbers when you submit the revised version.

Please note that when you prepare the revised manuscript have all the results (including tables and figures) in the result section. Tables and figures for pre-processing of the data can go to the data set and pre-processing section. Thus, I suggest the authors organize the paper as follows and insert the line numbers

Response: Thank you very much for this suggestion. We organize the paper according to the comment and insert the line numbers in the revised manuscript.

Comment: The data sets analyzed in this manuscript are only two time series! Your current conclusion section must be improved by adding the limitations of the study and future work.

Response: Thank you very much for this suggestion. We add the limitations and future work of the study in Section 4 Conclusion.

Comment: In the first paragraph of the Introduction please also include the following articles. The first one shows the application of a wavelet analysis software package (namely, the least-squares wavelet: LSWAVE) for streamflow and climate analysis and forecasting, and the second article talks about several AI techniques for climate forecasting:

https://doi.org/10.1016/j.ejrh.2021.100847

https://doi.org/10.3390/rs13163209

The least-squares wavelet analysis (LSWA) is a new method of analyzing non-stationary time series that may not be sampled at equally spaced time intervals and can consider observational uncertainties. LSWA has shown promise in a successful analysis of streamflow and climate time series that is a tool in the LSWAVE software (the first reference above). This can also be mentioned in the first paragraph on page 3 toward the end of the Introduction.

Response: Thank you very much for this suggestion. We add the two references in the revised manuscript. The added references are as follows:

  1. Ghaderpour, E.; Vujadinovic, T.; Hassan, Q.K. Application of the Least-Squares Wavelet software in hydrology: Athabasca River Basin. Journal of Hydrology: Regional Studies 2021, 36, 100847, doi:https://doi.org/10.1016/j.ejrh.2021.100847.
  2. Dewitte, S.; Cornelis, J.P.; Muller, R.; Munteanu, A. Artificial Intelligence Revolutionises Weather Forecast, Climate Monitoring and Decadal Prediction. Remote Sensing 2021, 13, doi:10.3390/rs13163209.

Comment: Equations (1) and (2) the variable “x” has a different format in Equation (2) and the first term on the right-hand side of Equation (1). Please check. It is not clear whether Equation (3) is derived from Equations (1) and (2). That should be clarified.

Response: Thank you very much for this suggestion. We modify the format of “x” in Equation (1), and Equation (3) is derived by adding Equations (1) and (2).

Comment: Equation (4) also uses “y” with two different formats. Please check the equations and correct them. The gradient symbol must also be defined.

Response: Thank you very much for this suggestion. We modify the format of “y” in Equation (4) and add the explanation of the gradient symbol.

Comment: Figure 1. The font size of all the variables should be the same and not too large.

Response: Thank you very much for this suggestion. We revise Figure 1 based on this comment. Please refer to the Section 2.2.2 BPNN for details.

Comment: Equation (5). Please also write “= 0, 1, …, n” and remove i(0,1) in the last line of page 4.

Response: Thank you very much for this suggestion. We replace “i(0,1)” with “i = 0, 1, …, n” .

Comment: Grammar. After Equation (5). It should be “formulas above” not “above formula”.

Response: Thank you very much for this suggestion. We replace “above formula” with “formulas above”.

Comment: The three lines below Equation (21): Figure 3 not Figure. 3. Also please never start a sentence with a variable or number. For example, instead of saying “x is the original signal”, please say “The original signal is shown by x”, etc.

Response: Thank you very much for this suggestion. We amend all inappropriate expressions in the paper based on this comment.

Comment: What about other symbols (names) in Figure 3? Please clearly describe them in the caption of Figure 3.

Response: Thank you very much for this suggestion. We add the explanation of other necessary symbols in Figure 3 to the revised manuscript. Please refer to Section 2.3.4 WPD Figure 6 for details.

Comment: I do not see anywhere in the text that the authors refer to Table 1. Please note that all the figures and tables must be clearly referred to in the body of the manuscript.

Response: Thank you very much for this comment. We add the description of Table 1 in Section 2.2 Data sets and pre-processing.

Comment: In the caption of Figures 5 and 6, please mention months since 1980. Is that true? Is it since 1980.

Response: Thank you very much for this comment. The abscissa of Figures 5 and 6 is replaced by the year.

Comment: Figures 13,14,15,16. Each figure should have two panels, the top panels are the current figures, but the bottom panels should be the residual time series (the result of each method subtracted from the observed series). This way the readers can easily see the differences. Also, I suggest display them as continuous lines rather than using markers. There are currently so many markers (dots) in Figures 13 and 15, making it very difficult to inspect the figures.

Response: Thank you very much for this comment. We improve the quality of Figures 13,14,15,16,17 based on the comment.

Comment: Abbreviations. All acronyms in the manuscript should be defined the first time they appear. The full name should come first followed by its abbreviation inside parentheses. Please be consistent. Also, please add an acronym table at the end of the manuscript listing all the acronyms used in the manuscript.

Response: Thank you very much for this comment. We amend all inappropriate expressions and add an acronym table in the appendix in the revised manuscript.

The above responses are our reply to the reviews about water-1389911. We look forward to hearing further information from you.

Sincerely yours

Round 2

Reviewer 1 Report

The title of this paper is “A comparison of BPNN, GMDH and ARIMA for monthly rainfall forecasting based on Wavelet Packet Decomposition”. The title of this paper is interesting. Many parts of this paper have been revised, but some contents need to be revised. This paper needs a revision for publication in Water. Therefore, I recommend a minor revision in this paper.

Author Response

Reply to questions or reviews given by Reviewer #1:

Comment: Figure2 and 3 needs an explanation.

Response: Thank you very much for your suggestion. In the revised manuscript, Figures 2 and 3 of the original manuscript are changed to Figures 4 and 5. We add explanations for Figures 4 and 5. Please refer to Figures 4 and 5 in the revised manuscript for details.

Comment: In Table 3, why are all critical values the same? The reason should be explained.

Response: Thank you very much for this comment. The critical value is a standard. For ADF test, when p-value < 0.05, the 5% test critical value is constant, which is equal to -3.420406. When t-statistical value is less than the critical value, it means that the sequence is stable.

The above responses are our reply to the reviews about water-1389911. We look forward to hearing further information from you.

Sincerely yours

Reviewer 3 Report

I would like to thank the authors for addressing my comments. The manuscript looks better now. I have a few minor comments:

Line 83. The reference here should be [2] not [45]. Please correct. Please check all the references to ensure that they are correctly referred to in the text.

Figure 1. It would be better to insert some of the latitudes and longitudes grids. Please either in the legend or in the caption mention that the elevation is in meters above the mean sea level. Also, please enlarge the station names in the figure. Please also ensure that all the figures are in high quality, e.g., at least 300 dpi resolution.

Equation (1). The first term in the right hand side of Equation (1) still says x (non-italic). Furthermore, the sum of Equations (1) and (2) is not Equation (3). Please clarify.

Table 3. Columns three (h) and four (pValue) could simply be removed. Instead, In the caption of Table 3, you can mention h is zero and pValue is zero for the Original and all WPDi.

Finally, please carefully proofread the article before publication. Checking the formulas, references, tables, etc.

Thank you for your contribution

Author Response

Reply to questions or reviews given by Reviewer #3:

Comment: Line 83. The reference here should be [2] not [45]. Please correct. Please check all the references to ensure that they are correctly referred to in the text.

Response: Thank you very much for this suggestion. We replace reference"[45]" with "[2]", and check all the references to ensure that they are correctly referred to in the paper.

Comment: It would be better to insert some of the latitudes and longitudes grids. Please either in the legend or in the caption mention that the elevation is in meters above the mean sea level. Also, please enlarge the station names in the figure. Please also ensure that all the figures are in high quality, e.g., at least 300 dpi resolution.

Response: Thank you very much for this suggestion. In the revised manuscript, the latitudes and longitudes grids have been added, and the station names in the figure1 have been enlarged. In addition, we improve the quality of Figure based on the comment.

Comment: Equation (1). The first term in the right-hand side of Equation (1) still says x (non-italic). Furthermore, the sum of Equations (1) and (2) is not Equation (3). Please clarify.

Response: Thank you very much for this suggestion. We modify the format of “x” in Equation (1). The ARMA model can be obtained by mixing AR and MA models, but they are not added directly. The white noise value of ARMA model is the same as that of AR and Ma model. Because  represents white noise and is the random fluctuation of values in time series, but these fluctuations will cancel each other and finally be 0.

Comment: Table 3. Columns three (h) and four (pValue) could simply be removed. Instead, In the caption of Table 3, you can mention h is zero and pValue is zero for the Original and all WPDi.

Response: Thank you very much for this suggestion. We modify Table 3 based on the comment in the revised manuscript.

The above responses are our reply to the reviews about water-1389911. We look forward to hearing further information from you.

Sincerely yours